# Variational Entropic Optimal Transport

**Roman Dyachenko** [1]  **Nikita Gushchin** [2 3]  **Kirill Sokolov** [4]  **Petr Mokrov** [2]  **Evgeny Burnaev** [2 3]
**Alexander Korotin** [2 3]

## Abstract

Entropic optimal transport (EOT) in continuous spaces with quadratic cost is a classical tool for solving the domain translation problem. In practice, recent approaches optimize a weak dual EOT objective depending on a single potential, but doing so is computationally not efficient due to the intractable log-partition term. Existing methods typically resolve this obstacle in one of two ways: by significantly restricting the transport family to obtain closed-form normalization (via Gaussian-mixture parameterizations), or by using general neural parameterizations that require simulation-based training procedures. We propose Variational Entropic Optimal Transport (VarEOT), based on an exact variational reformulation of the log-partition $\log \mathbb{E}[\exp(\cdot)]$ as a tractable minimization over an auxiliary log-normalizer. This yields a differentiable learning objective optimized with stochastic gradients and avoids the necessity of MCMC simulations during the training. We provide theoretical guarantees, including finite-sample generalization bounds and approximation results under universal function approximation. Experiments on synthetic data and unpaired image-to-image translation demonstrate competitive or improved translation quality, while comparisons within the solvers that use the same weak dual EOT objective support the benefit of the proposed optimization principle. The code for our solver can be found at https://github.com/DrEternity/VarEOT.

## 1. Introduction

Entropic Optimal Transport (EOT) with quadratic cost is well-established mathematical framework with strong theoretical properties that has found wide application in generative modeling and especially for unpaired domain translation. Despite this, the practical adoption of entropic transport methods has been limited by the lack of efficient and flexible algorithms. Existing approaches typically suffer from at least one of the following drawbacks: they are not amenable to simulation-free training, requiring costly sampling at each iteration (Mokrov et al., 2024); they require adversarial optimization (Gushchin et al., 2024b); they involve training a sequence of models rather than a single objective (Shi et al., 2023); they impose restrictive parametric forms on the transport plan (Korotin et al., 2024); or they turn out to be too sensitive to the entropic regularization strength (Daniels et al., 2021).

In our paper, we make a decisive step towards solving the drawbacks of existing EOT methods, and propose a novel simulation-free training solver based on an innovative variational reformulation of the weak dual EOT objective. We present the following **main contributions**:

1. **Variational dual objective** (§3.1). We derive an equivalent reformulation of the weak dual objective of entropic OT with quadratic cost in which the intractable log-partition term is replaced by an exact variational minimization over an auxiliary log-normalizer.

2. **Simulation-free training solver** (§3.2). Building on this reformulation, we propose a *simulation-free training* variational solver that jointly learns the dual potential and the auxiliary normalizer via neural parameterizations (no MCMC).

3. **Learning guarantees** (§3.3). We provide finite-sample learning guarantees for recovery of the entropic OT plan, decomposing error into estimation and approximation terms, and show vanishing approximation error under universal function approximation.

4. **Evaluation** (§5). We evaluate our solver on synthetic data and unpaired image-to-image translation. For the latter, we highlight gains by comparing against other solvers that optimize the same weak dual objective.

[1]Higher School of Economics, Moscow, Russia [2]Applied AI Institute, Moscow, Russia [3]AXXX, Russia [4]Lomonosov Moscow State University, Moscow, Russia. Correspondence to: Roman Dyachenko <rrdiachenko@edu.hse.ru>, Nikita Gushchin <i.nikita.gushchin@gmail.com>.

*Proceedings of the $43^{rd}$ International Conference on Machine Learning*, Seoul, South Korea. PMLR 306, 2026. Copyright 2026 by the author(s).

## 2. Background

This section provides the necessary background on entropic optimal transport and its optimization using weak dual reformulation. In §2.1, we recall the entropic optimal transport problem with quadratic cost, introduce its weak dual formulation, and describe the structure of the optimal transport plan induced by the optimal dual potential. In §2.2, we clarify our learning setup. In §2.3, we review existing solvers based on weak dual objective, highlighting two main paradigms: simulation-based optimization via implicit sampling proposed by the authors of EgNOT (Mokrov et al., 2024) and simulation-free methods based on restrictive parametric assumptions proposed by the authors of LightSB (Korotin et al., 2024). This discussion motivates the need for a dual formulation amenable to simulation-free training while remaining sufficiently expressive, which we address in subsequent §3 by proposing our novel VarEOT solver.

### 2.1. Entropic Optimal Transport with the quadratic cost

Let $p_0, p_1 \in \mathcal{P}_{\mathrm{ac}}(\mathbb{R}^D)$ be the set of absolutely continuous Borel probability measures on $\mathbb{R}^D$. Let $\Pi(p_0, p_1)$ denote the set of couplings (transport plans) on $\mathbb{R}^D \times \mathbb{R}^D$ with marginals $p_0$ and $p_1$. We write $\pi(x_0, x_1)$ for a plan density; $H$ is the differential entropy. The Entropic Optimal Transport (EOT) problem is given by:

$$\mathrm{EOT}_\varepsilon(p_0, p_1) \stackrel{\text{def}}{=} \min_{\pi \in \Pi(p_0, p_1)} \left\{ \underbrace{\mathop{\mathbb{E}}_{\pi(x_0, x_1)} \left[ \frac{\|x_0 - x_1\|^2}{2} \right]}_{\text{optimal transport term}} \right.$$

$$\left. \underbrace{-\varepsilon \int_{\mathbb{R}^D} H\big(\pi(\cdot \mid x_0)\big) p_0(x_0) dx_0}_{\text{entropic regularization term}} \right\}. \quad (1)$$

The entropic regularization term in (1) is due to (Mokrov et al., 2024). There are equivalent forms of the term (Cuturi, 2013; Léonard, 2014) which differ only by additive constants that do not affect the solution.

The EOT problem admits a unique minimizer $\pi^*$, referred to as the EOT plan. While the primal formulation (1) is conceptually appealing, it is computationally inconvenient, since enforcing the marginal constraints $\pi \in \Pi(p_0, p_1)$ requires optimizing over a complex set of probability measures that does not admit a straightforward parametrization.

**Weak dual form of EOT.** Objective (1) admits the following weak dual representation (Mokrov et al., 2024, Theorem 1):

$$\sup_{f \in L_1(\mathbb{R}^D)} \left\{ \underbrace{\mathop{\mathbb{E}}_{p_1(x_1)} f(x_1) - \varepsilon \mathop{\mathbb{E}}_{p_0(x_0)} \log Z(f, x_0)}_{\stackrel{\text{def}}{=} \mathcal{L}(f)} \right\}, \quad (2)$$

where the sup is taken over a function $f \in L_1(p_1)$ satisfying $f \equiv -\infty$ outside the $\mathrm{supp}(p_1)$ and

$$Z(f, x_0) \stackrel{\text{def}}{=} \int_{\mathbb{R}^D} \exp\left( \frac{f(x_1) - \frac{1}{2}\|x_0 - x_1\|^2}{\varepsilon} \right) dx_1 \quad (3)$$

is the partition function.

**Optimal transport plan.** Let $f^*$ be an optimizer of (2), the corresponding optimal transport plan $\pi^*$ (Mokrov et al., 2024, Theorem 1) can be recovered from it. By the disintegration with respect to the source marginal $p_0$ we have:

$$\pi^*(x_0, x_1) = \pi^*(x_1 \mid x_0) \, p_0(x_0), \quad (4)$$

Then the density of conditional distribution $\pi^*(\cdot \mid x_0)$ in (4) is directly defined by $f^*$:

$$\pi^*(x_1 \mid x_0) = \frac{1}{Z(f^*, x_0)} \exp\left( \frac{f^*(x_1) - \frac{1}{2}\|x_0 - x_1\|^2}{\varepsilon} \right). \quad (5)$$

### 2.2. Computational EOT setup

In practice, the source and target distributions, $p_0, p_1$, as well as the EOT objective (1), could be expressed and treated in different ways. To avoid possible misunderstanding, below we formalize our **practical learning setup**:

> We assume that source and target distributions $p_0$ and $p_1$ are accessible only by a limited number of i.i.d. empirical samples (datasets) $\{x_0^1, x_0^2, \dots x_0^N\} \sim p_0$; $\{x_1^1, x_1^2, \dots x_1^M\} \sim p_1$. Our aim is to approximate the optimal conditional plan $\pi^*(\cdot|x_0)$ (eq. (5)) between entire *distributions* $p_0$ and $p_1$. The recovered solution should provide the *out-of-sample* estimation, i.e., allow generating samples from $\pi^*(\cdot|x_0^{\mathrm{new}})$, where $x_0^{\mathrm{new}}$ is a new sample from $p_0$ which is not necessarily present in the train dataset.

This setup falls within **continuous** OT, in contrast to discrete OT (Cuturi, 2013; Peyré & Cuturi, 2019), which is designed to compute one-to-one or one-to-many *correspondence* directly between the collections of provided source and target samples. As a result, discrete OT approaches do not naturally accommodate the out-of-sample estimation demanded by continuous OT. In our manuscript, we focus exclusively on continuous OT approaches, treating discrete OT as a considerably different direction.

### 2.3. Existing Weak Dual Formulation Solvers

The practical optimization of problem (2) remains challenging due to the presence of the partition function $Z(f, x_0)$, which is, in general, intractable to compute exactly. Below we review two representative strategies to optimize the semidual objective: (i) general neural potentials with implicit sampling (EgNOT), and (ii) simulation-free objectives enabled by restrictive parametric transport families (LightSB).

**EgNOT solver.** The authors of EgNOT (Mokrov et al., 2024) solve (2) by parametrizing $f_\theta$ by a neural network and deriving the gradient of the weak dual objective $\mathcal{L}(f_\theta)$:

$$\nabla_\theta \mathcal{L}(f_\theta) = - \mathop{\mathbb{E}}_{p_0(x_0)} \left[ \mathop{\mathbb{E}}_{\pi_\theta(x_1|x_0)} [\nabla_\theta f_\theta(x_1)] \right] \\ + \mathop{\mathbb{E}}_{p_1(x_1)} [\nabla_\theta f_\theta(x_1)], \quad (6)$$

where $\pi_\theta(x_1|x_0)$ is given by:

$$\pi_\theta(x_1|x_0) = \frac{1}{Z(f_\theta, x_0)} \exp\left( \frac{f_\theta(x_1) - \frac{1}{2}\|x_0 - x_1\|^2}{\varepsilon} \right).$$

While flexible, **this approach is *not simulation-free***: each evaluation of the loss or its gradient requires sampling from the model distribution $\pi_\theta(x_1 \mid x_0)$, which is itself defined implicitly through the neural potential $f_\theta$. In practice, this sampling step is carried out using Markov chain Monte Carlo (MCMC) methods (Girolami & Calderhead, 2011; Hoffman et al., 2014; Samsonov et al., 2022), such as Langevin dynamics. However, MCMC-based sampling can be computationally expensive, sensitive to hyperparameters, and slow to mix, especially in high-dimensional settings.

**LightSB solver.** An alternative strategy is proposed in LightSB (Korotin et al., 2024). The authors introduce adjusted potential $v_\theta$ and parametrization:

$$\pi_\theta(x_1|x_0) = \frac{\exp\left(\langle x_0, x_1\rangle/\varepsilon\right) v_\theta(x_1)}{c_\theta(x_0)}, \quad (7)$$

where, $c_\theta(x_0) \overset{\text{def}}{=} \int_{\mathbb{R}^D} \exp\left(\langle x_0, x_1\rangle/\varepsilon\right) v_\theta(x_1) dx_1$ is the normalization. They then consider the optimization problem::

$$\min_{v_\theta} \left\{ \mathop{\mathbb{E}}_{p_0(x_0)} \log c_\theta(x_0) - \mathop{\mathbb{E}}_{p_1(x_1)} \log v_\theta(x_1) \right\}. \quad (8)$$

This problem is equivalent to the problem (2) under a different parametrization, specifically:

$$v_\theta(x_1) = \exp(-\frac{\|x_1\|^2}{2\varepsilon}) \exp(\frac{f_\theta(x_1)}{\varepsilon});$$
$$c_\theta(x_0) = \exp(\frac{\|x_0\|^2}{2\varepsilon}) Z(f_\theta, x_0).$$

To solve the problem in practice, the authors of LightSB circumvent the intractability of normalization $c_\theta(x_0)$ (equivalent of $Z(f, x_0)$, eq. (3)) by directly parameterizing the potential $v_\theta(x_1)$ in form of a Gaussian mixture:

$$v_\theta(x_1) = \sum_{k=1}^{K'} \alpha_k \mathcal{N}(x_1|r_k, \varepsilon S_k), \quad (9)$$

where $\theta \overset{\text{def}}{=} \{\alpha_k, r_k, S_k\}_{k=1}^{K'}$ are the parameters: $\alpha_k \geq 0$, $r_k \in \mathbb{R}^D$ and symmetric $0 \prec S_k \in \mathbb{R}^{D \times D}$. Such a parameterization restricts conditional density $\pi(x_1|x_0)$ to form:

$$\pi_\theta(x_1|x_0) = \frac{1}{c_\theta(x_0)} \sum_{k=1}^{K'} \widetilde{\alpha}_k(x_0) \mathcal{N}(x_1|r_k(x_0), \varepsilon S_k),$$

where $\widetilde{\alpha}(x_0)$ and $r_k(x_0)$ are some functions of $\{\alpha_k, r_k, S_k\}_{k=1}^{K'}$ and $c_\theta(x_0)$ is given in a closed form:

$$c_\theta(x_0) = \sum_{k=1}^{K'} \widetilde{\alpha}_k(x_0).$$

While this parameterization leads to a fully tractable and simulation-free objective, it **restricts the expressiveness of the method**, as it limits admissible transport plans to a rather narrow parametric family.

**Summary.** Existing weak dual solvers trade off between expressiveness and tractability: EgNOT supports flexible potentials but requires MCMC during training, while LightSB is simulation-free but restricts the conditional plan family. Below, we present our novel method, VarEOT, which takes the **best of two worlds** by enabling simulation-free training *without* restricting $\pi(x_1|x_0)$ to a narrow parametric family.

## 3. Variational Entropic Optimal Transport

In this section, we introduce our variational approach to entropic optimal transport. In §3.1, we derive a new variational dual formulation of the EOT objective that replaces the intractable log-partition function with a tractable variational upper bound, yielding a fully differentiable and simulation-free training loss. Building on this formulation, §3.2 presents a practical learning algorithm based on neural parameterization of the dual potential and the auxiliary variational function, together with a Langevin sampling procedure for generating transport samples. We further position our method relative to existing dual solvers and highlight its practical advantages. In §3.3 we conduct the analysis of our method from the perspectives of statistical learning theory (finite sampling learning guarantees and approximation with neural networks). All proofs are provided in Appendix A.

### 3.1. New Variational Dual Formulation of EOT

Our goal is to propose a weak dual solver that, unlike EgNOT and LightSB, does not require simulation during training and allows for expressive parameterization.

A key challenge in this setting is differentiating through the partition function $\log Z(f, x_0)$: as a logarithm of an expectation, it cannot be unbiasedly estimated from finite samples in a straightforward way. To overcome this difficulty, we adopt a variational approximation for the logarithm, which allows us to construct a tractable, differentiable surrogate for the dual objective. The details of this variational approach are formalized in the proposition below.

**Proposition 3.1** (Variational bound for the partition function). *The logarithm of partition function $\log Z(f, x_0)$ ad-*

*mits the variational upper bound:*

$$\log Z(f, x_0) \leq -1 + \xi(x_0) + \frac{D}{2}\log(2\pi\varepsilon) +$$
$$\mathop{\mathbb{E}}_{z \sim \mathcal{N}(0,I)} \left[ \exp\left( \frac{f(x_0 + \sqrt{\varepsilon}z)}{\varepsilon} - \xi(x_0) \right) \right], \quad (10)$$

*where $\xi : \mathbb{R}^D \to \mathbb{R}$ is an arbitrary integrable function. The upper bound is tight when*

$$\xi_f(x_0) = \log Z(f, x_0) - \frac{D}{2}\log(2\pi\varepsilon), \quad (11)$$

*i.e., at the (shifted by a constant) log partition function.*

Thanks to the obtained estimate, we can obtain tractable simulation-free loss:

**Theorem 3.2** (Variational dual form of EOT). *Let*

$$\mathcal{L}(f, \xi) \stackrel{def}{=} \varepsilon\left( 1 - \frac{D}{2}\log(2\pi\varepsilon) \right) +$$
$$\mathop{\mathbb{E}}_{x_1 \sim p_1}[f(x_1)] - \varepsilon \mathop{\mathbb{E}}_{x_0 \sim p_0}[\xi(x_0)] -$$
$$\varepsilon \mathop{\mathbb{E}}_{x_0 \sim p_0}\left[ \mathop{\mathbb{E}}_{z \sim \mathcal{N}(0,I)} \exp\left( \frac{f(x_0 + \sqrt{\varepsilon}z)}{\varepsilon} - \xi(x_0) \right) \right], \quad (12)$$

*Then the entropic optimal transport weak dual formulation admits the following variational form:*

$$\mathrm{EOT}_\varepsilon(p_0, p_1) = \sup_f \mathcal{L}(f) = \sup_{f,\xi} \mathcal{L}(f, \xi). \quad (13)$$

*The optimal solution $(f^*, \xi_f^*)$, where $\xi_f$ set by (11), recovers*

$$\pi^*(x_1 | x_0) = \frac{(2\pi\varepsilon)^{-\frac{D}{2}}}{\exp(\xi_f(x_0))} \exp\left( \frac{f^*(x_1) - \frac{1}{2}\|x_0 - x_1\|^2}{\varepsilon} \right)$$

This novel variational dual form allows us to overcome the original problem of estimation of log partition function $\log Z(f, x_0)$ in the weak dual form (2).

For convenience we define:

$$\pi^{f,\xi}(x_0, x_1) \stackrel{def}{=} \frac{(2\pi\varepsilon)^{-\frac{D}{2}} p_0(x_0)}{\exp(\xi(x_0))} \exp\left[ \frac{f(x_1) - \frac{1}{2}\|x_0 - x_1\|^2}{\varepsilon} \right];$$
$$\pi^f(x_0, x_1) \stackrel{def}{=} \pi^{f,\xi_f}(x_0, x_1),$$

where $\xi_f$ is due to (11). The following theorem establishes that the VarEOT optimality gap directly corresponds to the KL discrepancy between the recovered measure $\pi^{f,\xi}$ ($\pi^f$) and the EOT plan $\pi^*$.

**Theorem 3.3.** *For any $f \in L_1(p_1)$ and $\xi \in L_1(p_0)$,*

$$\varepsilon\mathrm{KL}\left(\pi^* \,\|\, \pi^f\right) \leq \varepsilon\mathrm{KL}\left(\pi^* \,\|\, \pi^{f,\xi}\right) = \mathcal{L}^* - \mathcal{L}(f, \xi), \quad (14)$$

*where $\mathrm{KL}(\cdot\|\cdot)$ denotes the Kullback–Leibler divergence between non-negative measures (see Definition A.1 in Appendix A), and $\mathcal{L}^*$ is the optimal value of the objective (12).*

Our Theorem 3.3 certifies that optimizing objective (12) directly enables us to approximate the ground truth EOT plan. Additionally, eq. (14) suggests that at inference stage it is better to use $\pi^f$, not $\pi^{f,\xi}$, which we exploit in our practical implementation, see the next section §3.2.

---

**Algorithm 1** Training procedure for Variational Entropic Optimal Transport (VarEOT)

**Input:** samples from distributions $p_0$ and $p_1$; entropy regularization parameter $\varepsilon > 0$; batch sizes $N_0, N_1$; number of noise samples $K$; potential network $\hat{f}_\theta : \mathbb{R}^D \to \mathbb{R}$; auxiliary network $\hat{\xi}_\psi : \mathbb{R}^D \to \mathbb{R}$.
**Output:** trained potential $\hat{f}_{\theta^*}$.
**for** each training iteration **do**
  Sample mini-batches $\{x_i^0\}_{i=1}^{N_0} \sim p_0$, $\{x_j^1\}_{j=1}^{N_1} \sim p_1$;
  Sample i.i.d. noise variables $z_{i,k} \sim \mathcal{N}(0, I)$ for $i = 1, \dots, N_0, k = 1, \dots, K$;
  Compute the empirical loss $\hat{\mathcal{L}}$ according to eq. (15);
  Update $\psi, \theta$ by using the gradients $\nabla_\psi\hat{\mathcal{L}}, \nabla_\theta\hat{\mathcal{L}}$;
**end for**

---

## 3.2. Practical Algorithm

**Training.** The variational formulation derived in Theorem 3.2 leads to a fully tractable and simulation-free training optimization objective, given in (12). In practice, we parametrize both the potential $f : \mathbb{R}^D \to \mathbb{R}$ and the auxiliary variational function $\xi : \mathbb{R}^D \to \mathbb{R}$ by neural networks, denoted by $f_\theta$ and $\xi_\psi$, respectively

The resulting training procedure consists of maximizing an empirical estimate of the variational dual loss with respect to both parameter sets $\theta$ and $\psi$. Crucially, both networks $f_\theta$ and $\xi_\psi$ are trained jointly in a single optimization loop, with no alternating or adversarial steps required. Given mini-batches of samples $\{x_i^0\}_{i=1}^{N_0} \sim p_0$ from the source distribution and $\{x_j^1\}_{j=1}^{N_1} \sim p_1$ from the target distribution, the expectations in eq. (12) are approximated using Monte Carlo sampling with i.i.d. Gaussian noise variables $z_{i,k} \sim \mathcal{N}(0, I)$, yielding the empirical loss, up to an additive constant in (12):

$$\widehat{\mathcal{L}}(f_\theta, \xi_\psi) \stackrel{def}{=} \frac{1}{N_1}\sum_{j=1}^{N_1} f_\theta(x_j^1) - \varepsilon\frac{1}{N_0}\sum_{i=1}^{N_0} \xi_\psi(x_i^0) -$$
$$\frac{\varepsilon}{N_0 K}\sum_{i,k=1}^{N_0,K} \exp\left( \frac{f_\theta(x_i^0 + \sqrt{\varepsilon}\,z_{i,k})}{\varepsilon} - \xi_\psi(x_i^0) \right). \quad (15)$$

Importantly, **this approximation *does not require sampling from the model distribution itself***, in contrast to energy-based approaches such as EgNOT. The complete training procedure is summarized in Algorithm 1. To mitigate numerical instability arising from the exponential terms in (15), we clip their values prior to gradient computation and additionally apply gradient clipping during training; see Appendix B for details.

**Inference.** Following the hint in equation (14), after the training we only use the potential function $\hat{f}_\theta$ for inference,

**Algorithm 2** Langevin sampling from the VarEOT conditional distribution

---

**Input:** source sample $x_0 \sim p_0$;
trained potential network $\hat{f}_\theta$;
entropy regularization $\varepsilon > 0$;
number of Langevin steps $S$; step size $\eta > 0$.
**Output:** sample $x_1 \sim \pi(\cdot \mid x_0)$.

Initialize $x_1^{(0)} = x_0$;
**for** $s = 1$ **to** $S$ **do**
    Sample $z^{(s)} \sim \mathcal{N}(0, I)$;
    $h = \frac{1}{\varepsilon}\left( \nabla_{x_1} \hat{f}_\theta(x_1^{(s-1)}) - \left(x_1^{(s-1)} - x_0\right) \right)$;
    Update:
$$x_1^{(s)} \leftarrow x_1^{(s-1)} + \eta h + \sqrt{2\eta}\, z^{(s)};$$
**end for**
**Return** $x_1^{(S)}$

---

whereas the function $\hat{\xi}_\psi$ is not used. The corresponding entropic optimal transport plan is implicitly defined via the conditional distribution (5). To generate samples from this conditional distribution, we employ Langevin dynamics targeting the unnormalized density

$$\pi(x_1 \mid x_0) \propto \exp\left( \frac{\hat{f}_\theta(x_1) - \frac{1}{2}\|x_1 - x_0\|^2}{\varepsilon} \right).$$

The resulting sampling procedure is detailed in Algorithm 2. In all experiments, we find that $10^1$-$10^3$ Langevin steps with an appropriate step size suffice for high-quality samples.

To highlight the practical advantages of VarEOT, we compare it with two representative dual-solver approaches: EgNOT (Mokrov et al., 2024) and LightSB (Korotin et al., 2024) (cf. §2.3). Table 1 summarizes their properties in terms of simulation-free training and structural constraints on the transport plan.

*Table 1.* Comparison of dual solvers for entropic OT. "Simulation-free" indicates whether the method avoids sampling from the model during training. "Not Restricted parameterization" indicates whether the conditional plan is not restricted to a predefined family (e.g., Gaussian mixtures).

| Method | Simulation-free training | Not restricted parameterization |
|---|:---:|:---:|
| EgNOT | ✗ | ✓ |
| LightSB | ✓ | ✗ |
| VarEOT (**ours**) | ✓ | ✓ |

VarEOT combines the best of both worlds: it is *simulation-free training*, unlike EgNOT, and it imposes no *restricted parameterization* on the conditional plan, unlike LightSB.

### 3.3. Finite Sample Learning Guarantees

In this subsection we quantify the discrepancy between the transport plan recovered by VarEOT and the ground truth EOT solution. Our method does not have access to $(p_0, p_1)$ explicitly; instead it works with finite samples (see our computational setup §2.2) and with potentials restricted to parametric classes. This naturally introduces several sources of error in practice: *finite-sample error* (we only observe i.i.d. datasets of sizes $N$ and $M$ from $p_0$ and $p_1$); *function class restriction* (we optimize over a class of neural networks rather than over all continuous potentials); *optimization error* (stochastic training, finite-time optimization). Our rigorous theoretical analysis below focuses on the first two items. The third item depends on the chosen optimizer and sampling scheme and is treated separately in practice.

Throughout, the function classes $\mathcal{F}$ and $\Xi$ are understood as classes of feedforward neural networks with biases, prescribed depth and width, uniformly bounded layer norms, and clipped outputs. Here, clipped outputs mean that the final network output is truncated to lie in a fixed bounded interval. We additionally assume that the hidden activation $\sigma$ is $L_\sigma$-Lipschitz, non-polynomial and satisfies $\sigma(0) = 0$. Thus,

$$\mathcal{F} := \mathsf{NN}_\theta^\sigma(D, 1), \quad \Xi := \mathsf{NN}_\eta^\sigma(D, 1), \quad (16)$$

We begin with a decomposition of the excess KL discrepancy between the ground-truth EOT optimizer $\pi^*$ and the learned plan $\pi^{\hat{f}}$ (cf. (14)), where

$$(\hat{f}, \hat{\xi}) = \arg\max_{f \in \mathcal{F},\, \xi \in \Xi} \widehat{\mathcal{L}}(f, \xi).$$

Here $\xi$ enters only through the training objective, whereas the transport plan used at inference time is determined exclusively by the learned potential $\hat{f}$; accordingly, the final discrepancy is measured in terms of $\pi^{\hat{f}}$, cf. §3.2.

**Proposition 3.4.** *The following bound holds:*

$$\varepsilon \mathbb{E}\Big[\mathrm{KL}\big(\pi^* \,\|\, \pi^{\hat{f}}\big)\Big] \leq \underbrace{2\mathbb{E} \sup_{f \in \mathcal{F},\, \xi \in \Xi} \Big|\mathcal{L}(f, \xi) - \widehat{\mathcal{L}}(f, \xi)\Big|}_{\text{Estimation error}} +$$
$$\underbrace{\mathcal{L}^* - \sup_{f \in \mathcal{F},\, \xi \in \Xi} \mathcal{L}(f, \xi)}_{\text{Approximation error}}, \quad (17)$$

*where the expectations are taken w.r.t. the random realization of the datasets.*

Proposition 3.4 separates the excess error into a statistical term (estimation error) and a representational term (approximation error). The first measures the uniform discrepancy between the empirical and population objectives over the prescribed neural-network classes, while the second quantifies the gap from restricting the optimization to these classes.

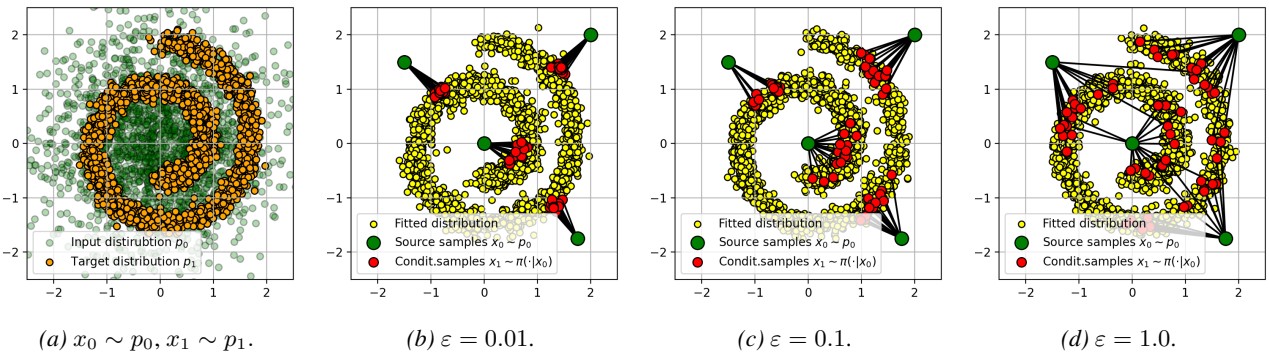

*Figure 1.* Optimal plan learned with VarEOT **(ours)** in *Gaussian→Swiss roll* example.

The next theorem controls the statistical term. It shows that, for fixed network classes, the estimation error decays at the usual sublinear rate as the number of samples increases. Importantly, decay rates do not depend on the dimension $D$.

**Theorem 3.5** (Bound on estimation error). *Assume that the classes $\mathcal{F}$ and $\Xi$ are defined by* (16)*, and that $p_0$ and $p_1$ have compact supports. Then*

$$\mathbb{E} \sup_{f \in \mathcal{F}, \, \xi \in \Xi} \left| \mathcal{L}(f, \xi) - \widehat{\mathcal{L}}(f, \xi) \right| \le$$
$$\le O\left( N^{-\frac{1}{2}} \right) + O\left( M^{-\frac{1}{2}} \right) + O\left( K^{-\frac{1}{2}} \right) \quad (18)$$

*where the hidden constants depend on $\varepsilon$, the dimension $D$, the complexity parameters of the classes $\mathcal{F}$ and $\Xi$ (in particular, the layer-norm bounds, clipping level, width, and depth), and the support diameters of $p_0$ and $p_1$. Expectation is taken w.r.t. random realizations of the dataset.*

We next turn to the approximation term. The following statement asserts that it can be made arbitrarily small by choosing sufficiently expressive neural-network classes.

**Theorem 3.6** (Vanishing of approximation error). *Let $p_0$ and $p_1$ have compact supports, then for any $\delta > 0$, there exist classes $\mathcal{F}$ and $\Xi$ of the form* (16) *such that*

$$\mathcal{L}^* - \sup_{f \in \mathcal{F}, \, \xi \in \Xi} \mathcal{L}(f, \xi) < \delta. \quad (19)$$

Thus, the bias introduced by the neural-network parameterization can be driven arbitrarily close to zero by enlarging the expressive power of the admissible classes.

**Summary.** Proposition 3.4, together with Theorems 3.5 and 3.6, yields the standard learnability picture for EOT plan recovery. For fixed neural-network classes, the estimation gap vanishes as the sample sizes grow. Meanwhile, by increasing the expressiveness of the classes, the approximation error can be made arbitrarily small. Consequently, with sufficiently many samples and sufficiently expressive neural-network parameterizations, the learned plan $\pi^{\hat{f}}$ approaches the EOT solution in excess KL discrepancy.

**Relation to prior work.** Related error decompositions appear in (Mokrov et al., 2024; Kolesov et al., 2024). While (Mokrov et al., 2024) does not provide a fully detailed breakdown of the individual error terms for EgNOT, (Kolesov et al., 2024) develops a finer-grained statistical analysis but for a different objective (entropic barycenters). We follow a similar template, yet provide a detailed analysis for our more elaborate variationally derived VarEOT functional.

# 4. Related Work

In this section, we review the existing continuous EOT approaches, and briefly compare them with our solver. For clarity, we introduce a taxonomy of the existing approaches below, based on the utilized mathematical framework.

**Weak dual EOT** (a.k.a. *semi*-dual EOT) methods (Mokrov et al., 2024; Korotin et al., 2024) optimize the objective eq. (2). They are the most similar to our VarEOT in terms of methodology. The detailed discussion of the approaches is given in §2.3. Crucially, our proposed method successfully overcomes the limitations of existing weak dual approaches while retaining their advantages.

**Dual EOT** (Genevay et al., 2016; Seguy et al., 2018; Daniels et al., 2021) exploit an alternative dual form of the EOT. These methods simultaneously optimize a *pair* of dual potentials $(u, v)$ in a $\max \max$ optimization procedure known as the Sinkhorn algorithm. While bearing certain resemblance to our VarEOT (e.g., simulation-free training), dual EOT approaches have pitfalls underscoring their performance:

- Dual potentials $(u, v)$ do not fully recover the (conditional) EOT $\pi^*(\cdot|x_0)$. In particular, an *auxiliary* score-based model for the target distribution $p_1$ is required to enable sampling from $\pi^*(\cdot|x_0)$ (Daniels et al., 2021). Otherwise, there are only some heuristics (e.g., barycentric projection) that allow the recovered solution to be cast as a *generative* model capable of data-to-data translation (Genevay et al., 2016; Seguy et al., 2018).

| Setup | Solver type | DIM / Solver | 50 | 100 | 1000 |
|---|---|---|---|---|---|
| Discrete EOT | Sinkhorn | (Cuturi, 2013) [1 GPU V100]* | 2.34 (90 s) | 2.24 (2.5 m) | 1.864 (9 m) |
| Continuous EOT | Langevin-based | (Mokrov et al., 2024) [1 GPU V100]* | $2.39 \pm 0.06$ (19 m) | $2.32 \pm 0.15$ (19 m) | $1.46 \pm 0.20$ (15 m) |
| Continuous EOT | Minimax | (Gushchin et al., 2023) [1 GPU V100]* | $2.44 \pm 0.13$ (43 m) | $2.24 \pm 0.13$ (45 m) | $1.32 \pm 0.06$ (71 m) |
| Continuous EOT | IPF | (Vargas et al., 2021) [1 GPU V100]* | $3.14 \pm 0.27$ (8 m) | $2.86 \pm 0.26$ (8 m) | $2.05 \pm 0.19$ (11 m) |
| Continuous EOT | Variational | **VarEOT (ours)** [GTX 1080] | $\mathbf{2.45 \pm 0.08}$ (3 m) | $\mathbf{2.31 \pm 0.15}$ (3 m) | $\mathbf{1.47 \pm 0.11}$ (14.8 m) |

*Table 2.* Energy distance (averaged for two setups and 5 random seeds) on the MSCI dataset along with 95%-confidence interval ($\pm$ intervals) and average training times (s — seconds, m — minutes). Results marked with * are taken from (Korotin et al., 2024).

- Due to optimization peculiarities, dual EOT approaches tend to be unstable under small regularization coefficient $\varepsilon$. This behavior is reported in (Daniels et al., 2021, §5.1) and supported by (Mokrov et al., 2024, §4.2, §C.2).

Overall, VarEOT provides a more user-friendly and production-ready framework with less engineering overhead required to adapt the method to applications.

**Schrödinger bridge (SB)** methods (De Bortoli et al., 2021; Vargas et al., 2021; Chen et al., 2022; Gushchin et al., 2023; Shi et al., 2023; Tong et al., 2024; Gushchin et al., 2024b; De Bortoli et al., 2024; Gushchin et al., 2024a) cast the EOT in a dynamic manner and recover a solution in the form of a stochastic differential equation with learnable drift, with the endpoints given by source $p_0$ and target $p_1$ distributions. The underlying principles of SB-based approaches are different, e.g., adversarial optimization, gradual iterative markovian/proportional fitting procedure. However, compared to our method, **majority** of them are not simulation-free at training. The exceptions are (Tong et al., 2024) (based on mini-batch EOT approximation, which may not be accurate) and (Gushchin et al., 2024a) (similar to (Korotin et al., 2024) uses restricted Gaussian mixture approximation).

# 5. Experimental Illustrations

We evaluate VarEOT on both synthetic and real-world data. §5.1 presents illustrative two-dimensional experiments demonstrating the effect of the regularization parameter $\varepsilon$ on the learned transport plan. §5.2 benchmarks VarEOT on high-dimensional biological single-cell data from the MSCI dataset, comparing against EOT/SB solvers. §5.3 comprehensively evaluates on unpaired image-to-image translation using the standard ALAE latent space protocol, comparing VarEOT against related entropic transport baselines across multiple translation tasks and regularization regimes. Technical details, including architecture specifications and hyperparameters, are provided in Appendix B.

## 5.1. Two-dimensional Examples

We begin with a standard illustrative two-dimensional experiment that provides intuition into the role of the entropy

regularization parameter $\varepsilon$ in shaping the learned conditional transport plan $\pi_\theta(x_1 \mid x_0)$. Specifically, we consider a setting in which a Gaussian source distribution is transported to a Swiss Roll target distribution. We apply our method for $\varepsilon \in \{10^{-2}, 10^{-1}, 10^0\}$, and visualize the resulting transport maps in Fig. 1.

For small values of $\varepsilon$, the learned transport is nearly deterministic, with little variation in sampling endpoints. As $\varepsilon$ increases, the transport becomes progressively more stochastic: the endpoints diversify, and the conditional distributions $\pi_\theta(x_1 \mid x_0)$ spread over a wider region of the target space.

## 5.2. Single Cell Data

One of the important applications of EOT is the analysis of biological single-cell data (Koshizuka & Sato, 2023; Bunne et al., 2023; 2022). We evaluate VarEOT on the high-dimensional MSCI dataset (Kaggle competition "Open Problems — Multimodal Single-Cell Integration") first used in (Tong et al., 2024). The dataset consists of single-cell data from four human donors at 4 time points (days 2, 3, 4, and 7). We solve the EOT problem between distribution pairs at days $2 \rightarrow 4$ and $3 \rightarrow 7$, and evaluate how well the solvers recover the intermediate distributions at days 3 and 4 respectively. We consider PCA projections with DIM $= 50, 100, 1000$ components and report energy distance (Rizzo & Székely, 2016) (ED) in Table 2. For VarEOT, the experimental setup (architecture, number of iterations, $\varepsilon$, and learning rate) follows the EgNOT configuration from (Korotin et al., 2024) with $K{=}128$ particles; we refer the reader to Appendix D.4 of (Korotin et al., 2024) for full details. VarEOT achieves quality comparable to Langevin-based and Minimax solvers for DIM $= 50$ and 100, while being 5–15$\times$ faster (3 m vs. 19–45 m) on less powerful hardware (GTX 1080 vs. V100). For DIM $= 1000$, VarEOT matches the quality of the Langevin-based solver (1.47 vs. 1.46) with comparable training time (14.8 m vs. 15 m). The details of preprocessing, hyperparameters and baselines are in Appendix B.

## 5.3. Unpaired Image-to-Image Translation

Unpaired image-to-image translation is a standard benchmark in the entropic optimal transport (EOT) and

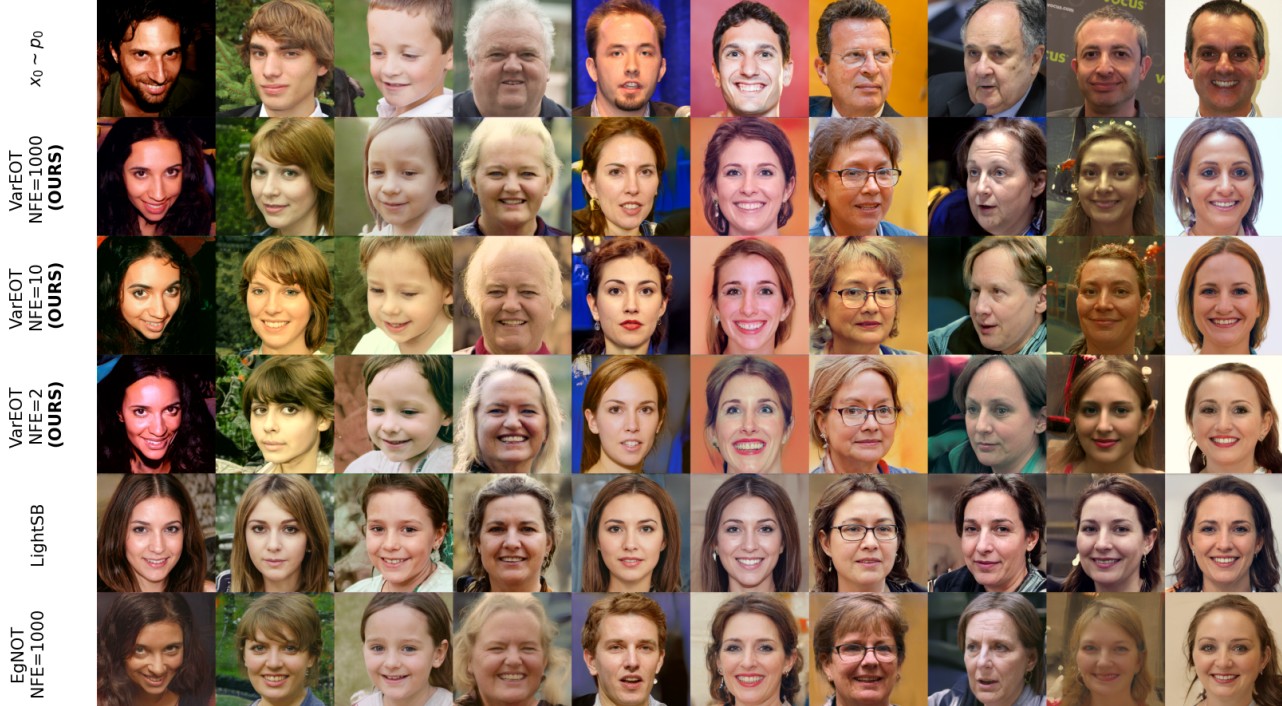

*Figure 2.* Qualitative comparison for *Man → Woman* translation with $\varepsilon = 1.0$. From top to bottom: input samples, VarEOT (ours), LightSB, and EgNOT. Input images are selected from the test set: we take the first 300 samples and rank them by encoder-decoder reconstruction quality (LPIPS), displaying the top-ranked examples.

Schrödinger Bridge (SB) literature (Zhu et al., 2017; Daniels et al., 2021; Chen et al., 2021), as it naturally combines high-dimensional data and the need for stochastic transport plans. In particular, translation in the latent space of pretrained autoencoders has become a popular and convenient setup for comparing entropic transport methods (Korotin et al., 2024; Mokrov et al., 2024; Kornilov et al., 2024).

We follow the widely adopted ALAE protocol (Korotin et al., 2024; Theodoropoulos et al., 2024; Gushchin et al., 2024a; Kornilov et al., 2024; Gazdieva et al., 2024; Han et al., 2025) based on ALAE autoencoder (Pidhorskyi et al., 2020). Specifically, we use a pretrained ALAE autoencoder trained on the full $1024 \times 1024$ FFHQ dataset (Karras et al., 2019), which contains approximately 70K human face images. The first 60K images are used for training and are split into (*male*, *female*) and (*child*, *adult*) subsets. Using the fixed ALAE encoder, we extract 512-dimensional latent representations $\{z_0^n = \mathrm{Enc}(x_0^n)\}_{n=1}^N$ and $\{z_1^m = \mathrm{Enc}(x_1^m)\}_{m=1}^M$ corresponding to the source and target domains. We consider 4 setups: male to female (M→F), female to male (F→M), adult to child (A→C), and child to adult (C→A).

**Training.** Given unpaired samples from the two latent distributions, we learn a latent entropic optimal transport plan $\pi_\theta(z_1 \mid z_0)$ using our variational formulation. The model is trained entirely in latent space and does not require paired data or image-level supervision.

**Inference.** To translate a previously unseen *image* $x_0^{\mathrm{new}}$ (from the remaining 10K test images), we **(i)** encode it as $z_0^{\mathrm{new}} = \mathrm{Enc}(x_0^{\mathrm{new}})$, **(ii)** sample $z_1 \sim \pi_\theta(z_1 \mid z_0^{\mathrm{new}})$ using Langevin dynamics, and **(iii)** decode $x_1 = \mathrm{Dec}(z_1)$.

**Evaluation Protocol.** We evaluate the quality of generated samples using two metrics: Fréchet Inception Distance (FID) (Heusel et al., 2017) and Learned Perceptual Image Patch Similarity (LPIPS) (Zhang et al., 2018). FID is computed between the set of test translated images $\{x_1^n = \mathrm{Dec}(z_1^n)\}_{n=1}^N$, where $z_1^n \sim \pi_\theta(z_1 \mid z_0^n)$, and the set of ALAE-reconstructed target images $\{\tilde{x}_1^m = \mathrm{Dec}(\mathrm{Enc}(x_1^m))\}_{m=1}^M$. LPIPS is measured between the source $x_0$ and the translated $x_1 = \mathrm{Dec}(z_1)$, where $z_1 \sim \pi_\theta(z_1 \mid \mathrm{Enc}(x_0))$. It measures the input-output similarity.

**Results.** We comprehensively evaluate VarEOT across all four translation setups: M→F, F→M, A→C, and C→A. Figure 2 shows qualitative results for the M→F setup with $\varepsilon = 1.0$. We report translations obtained using different numbers of Langevin steps at inference time (NFE $= 2, 10, 1000$), demonstrating that VarEOT produces visually plausible and stable results even with a very small number of sampling steps. Additional results for alternative translation directions (F→M, A→C, C→A) are provided in Appendix C (Figures 3, 4, and 5), confirming consistent behavior across all setups. Quantitative evaluation using FID and LPIPS metrics is summarized in Ta-

| $\varepsilon$ | Task | FID ↓ | | | | | | LPIPS ↓ | | | | | |
|---|---|---|---|---|---|---|---|---|---|---|---|---|---|
| | | LightSB | EgNOT | | VarEOT (ours) | | | LightSB | EgNOT | | VarEOT (ours) | | |
| | | | NFE=10 | NFE=1000 | NFE=2 | NFE=10 | NFE=1000 | | NFE=10 | NFE=1000 | NFE=2 | NFE=10 | NFE=1000 |
| 0.1 | M→F | 8.77 | 13.4 | **7.63** | 181.44 | 11.70 | 12.74 | 0.582 | 0.6 | 0.585 | 0.872 | 0.580 | **0.570** |
| | F→M | 10.87 | 14.9 | **6.33** | 248.76 | 17.00 | 13.98 | 0.597 | 0.603 | 0.598 | 0.833 | 0.586 | **0.578** |
| | A→C | 15.42 | 40.28 | **10.52** | 247.81 | 50.35 | 17.33 | 0.577 | 0.586 | 0.588 | 0.839 | **0.545** | 0.577 |
| | C→A | 13.46 | 18.47 | **10.4** | 253.59 | 21.48 | 20.13 | 0.594 | 0.602 | 0.608 | 0.865 | 0.591 | **0.590** |
| 0.5 | M→F | 19.09 | 35.4 | 14.07 | **9.04** | 9.50 | 9.77 | 0.611 | 0.718 | 0.594 | 0.619 | 0.600 | **0.598** |
| | F→M | 25.81 | 51.73 | 11.66 | 20.01 | 10.02 | **9.94** | 0.628 | 0.695 | 0.613 | **0.609** | 0.619 | 0.616 |
| | A→C | 22.30 | 74.25 | **18** | 78.85 | 26.38 | 25.99 | 0.608 | 0.702 | 0.603 | **0.593** | 0.617 | 0.613 |
| | C→A | 22.70 | 40.38 | 16.95 | 17.83 | **16.55** | 16.74 | **0.614** | 0.7 | 0.629 | 0.632 | 0.616 | **0.614** |
| 1.0 | M→F | 22.63 | 39.41 | 17.58 | 10.38 | **9.59** | 9.71 | 0.637 | 0.749 | 0.634 | 0.633 | 0.613 | **0.610** |
| | F→M | 20.97 | 66.64 | 26.21 | 15.21 | 10.52 | **10.28** | 0.649 | 0.72 | 0.634 | 0.654 | 0.633 | **0.630** |
| | A→C | 24.43 | 60.57 | 28.45 | 53.39 | **16.60** | 16.83 | 0.634 | 0.724 | 0.623 | 0.700 | 0.610 | **0.608** |
| | C→A | 24.87 | 51.48 | 23.68 | **15.73** | 16.55 | 16.70 | 0.637 | 0.733 | 0.661 | 0.638 | 0.623 | **0.619** |
| 10.0 | M→F | 31.85 | 37.61 | 26.68 | **21.84** | 21.97 | 22.38 | 0.680 | 0.755 | 0.735 | 0.625 | 0.621 | **0.620** |
| | F→M | 34.47 | 63.73 | 36 | **25.55** | 28.84 | 29.24 | 0.693 | 0.758 | 0.734 | 0.637 | 0.632 | **0.630** |
| | A→C | **31.85** | 92.6 | 64.85 | 49.84 | 50.72 | 51.20 | 0.680 | 0.767 | 0.763 | 0.619 | **0.615** | 0.615 |
| | C→A | 35.35 | 51.74 | 35.53 | **31.01** | 32.84 | 34.07 | 0.683 | 0.748 | 0.741 | 0.639 | 0.636 | **0.633** |

*Table 3.* Quantitative comparison of unpaired image-to-image translation methods in the ALAE latent space. We report FID and LPIPS (lower is better) for four translation tasks (M→F, F→M, A→C, C→A) across different values of the entropic regularization parameter $\varepsilon$. VarEOT results are shown for different numbers of Langevin inference steps (NFE = 2, 10, 1000). For each row, the best-performing method is highlighted in **bold**.

ble 3, covering a wide range of regularization parameters $\varepsilon \in \{0.1, 0.5, 1.0, 10.0\}$ and all four translation directions. We compare VarEOT against closely related entropic transport baselines: LightSB and EgNOT, showing competitive or improved performance across all considered regimes.

A qualitative analysis of the effect of the regularization parameter $\varepsilon$ on sample diversity is provided in Figure 6, illustrating how varying $\varepsilon$ controls the trade-off between transport cost and output stochasticity. Finally, Figure 7 provides an in-depth study of FID performance as a function of the Langevin step size and the number of inference steps.

We additionally conduct a wall-clock time analysis as a function of the number of particles $K$. Table 4 shows that VarEOT training time scales with $K$: larger $K$ leads to longer training, but also to better FID. Importantly, VarEOT achieves superior FID compared to both LightSB and Eg-NOT across all considered values of $K$, while requiring significantly less training time than EgNOT (e.g., 131s vs. 1692s for better quality). We attribute this to our simulation-free training objective, which avoids costly trajectory simulation required by EgNOT.

## 6. Discussion

**Potential impact.** By introducing an exact variational reformulation of the weak dual EOT objective, VarEOT makes a step toward more efficient entropic transport algorithms that avoid the key limitations of existing methods, such as simulation-based training, adversarial objectives, and restrictive parametric assumptions.

**Limitations.** While VarEOT enables simulation-free training, several limitations remain. First, sampling at inference time via Langevin dynamics is still required, making generation quality dependent on step size and step count. Second, the exponential terms in the objective may cause numerical instability during training. Third, the quality of optimization is inherently tied to the expressiveness of the auxiliary network $\xi$, and the choice of its architecture remains a practical question; we note that in our experiments a standard MLP for $\xi$ proved sufficient. Finally, the current framework is specialized to the quadratic cost (Gaussian kernel); extending to general costs is a natural direction for future work.

## Acknowledgements

The work was supported by the grant for research centers in the field of AI provided by the Ministry of Economic Development of the Russian Federation in accordance with the agreement 000000C313925P4F0002 and the agreement №139-10-2025-033.

## Impact Statement

This paper presents work whose goal is to advance the field of Machine Learning. There are many potential societal consequences of our work, none which we feel must be specifically highlighted here.

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

## A. Proofs

Throughout the appendix, we will use the shorthand $Z_f := Z(f, \cdot)$.

**Definition A.1.** Let $\mu$ and $\nu$ be non-negative measures on $\mathbb{R}^d$, and $\mu$ absolutely continuous w.r.t. $\nu$. The Kullback-Leibler divergence is defined by

$$\mathrm{KL}(\mu \| \nu) := \int \left[ \frac{d\mu}{d\nu} \log \left( \frac{d\mu}{d\nu} \right) - \frac{d\mu}{d\nu} + 1 \right] d\nu.$$

**Definition A.2** (Empirical Rademacher complexity)**.** Let $S = (x_1, \ldots, x_N) \in X^N$ be a fixed sample, and let $\mathcal{H}$ be a class of measurable functions $h : X \to \mathbb{R}$. Its empirical Rademacher complexity on $S$ is defined by

$$\widehat{\mathfrak{R}}_S(\mathcal{H}) := \frac{1}{N} \mathbb{E}_\sigma \left[ \sup_{h \in \mathcal{H}} \sum_{i=1}^N \sigma_i h(x_i) \right],$$

where $\sigma_1, \ldots, \sigma_N$ are independent Rademacher random variables.

**Definition A.3** (Rademacher complexity)**.** Let $p$ be a probability measure on $X \subseteq \mathbb{R}^D$, and let $\mathcal{H}$ be a class of measurable functions $h : X \to \mathbb{R}$. Its Rademacher complexity is

$$\mathfrak{R}_N(\mathcal{H}, p) := \mathbb{E}_{S \sim p^N} \widehat{\mathfrak{R}}_S(\mathcal{H}) = \frac{1}{N} \mathbb{E}_{x, \sigma} \left[ \sup_{h \in \mathcal{H}} \sum_{i=1}^N \sigma_i h(x_i) \right],$$

where $x_1, \ldots, x_N$ are i.i.d. samples from $p$.

The following symmetrization bound is standard; see, e.g., (Shalev-Shwartz & Ben-David, 2014, Lemma 26.2).

**Lemma A.4** (Representativeness estimation)**.** *Let $\mathcal{H}$ be a class of measurable functions $h : X \to \mathbb{R}$ such that $\mathcal{H} \subset L^1(p)$. Then for i.i.d. samples $x_1, \ldots, x_N \sim p$,*

$$\mathbb{E} \sup_{h \in \mathcal{H}} \left| \mathbb{E}_p[h] - \frac{1}{N} \sum_{i=1}^N h(x_i) \right| \leq 2 \mathfrak{R}_N(\mathcal{H}, p).$$

We will use the vector contraction inequality of (Maurer, 2016, Corollary 4).

**Lemma A.5** (Vector contraction)**.** *Let $S = (x_1, \ldots, x_N)$ be a fixed sample, let $\mathcal{F}$ be a class of functions $f = (f_1, \ldots, f_K) : X \to \mathbb{R}^K$, and let $h_i : \mathbb{R}^K \to \mathbb{R}$ be $L$-Lipschitz with respect to the Euclidean norm. Then*

$$\mathbb{E}_\sigma \left[ \sup_{f \in \mathcal{F}} \sum_{i=1}^N \sigma_i \, h_i(f(x_i)) \right] \leq \sqrt{2} \, L \, \mathbb{E}_{\sigma_{ik}} \left[ \sup_{f \in \mathcal{F}} \sum_{i=1}^N \sum_{k=1}^K \sigma_{ik} f_k(x_i) \right],$$

*where $\{\sigma_{ik}\}_{1 \leq i \leq N, \, 1 \leq k \leq K}$ is an independent doubly indexed Rademacher family.*

**Lemma A.6.** *Let $\mathcal{U}, \mathcal{V}$ be classes of measurable functions from $X$ to $\mathbb{R}$, and let a function $\psi$ satisfy*

$$|\psi(u, v) - \psi(u', v')| \leq L_u |u - u'| + L_v |v - v'|$$

*for any pairs $(u, v), (u', v')$ from the range of values $\mathcal{U}, \mathcal{V}$. Then*

$$\mathfrak{R}_N \big( \{ \psi(u(\cdot), v(\cdot)) : u \in \mathcal{U}, \, v \in \mathcal{V} \}, \, p \big) \leq 2L_u \mathfrak{R}_N(\mathcal{U}, p) + 2L_v \mathfrak{R}_N(\mathcal{V}, p).$$

*Proof.* Denote

$$\mathcal{G} := \{ \psi(u(\cdot), v(\cdot)) : u \in \mathcal{U}, \, v \in \mathcal{V} \}.$$

Fix a sample $S = (x_1, \ldots, x_N)$ and define the vector-valued class

$$\mathcal{F} := \{ x \mapsto (L_u u(x), L_v v(x)) : u \in \mathcal{U}, \, v \in \mathcal{V} \}.$$

Also define,

$$h(a, b) := \psi(a/L_u, b/L_v)$$

For all feasible pairs $(a, b), (a', b')$, we have

$$|h_i(a, b) - h_i(a', b')| = |\psi(a/L_u, b/L_v) - \psi(a'/L_u, b'/L_v)|$$

$$\leq L_u \left| \frac{a - a'}{L_u} \right| + L_v \left| \frac{b - b'}{L_v} \right| = |a - a'| + |b - b'| \leq \sqrt{2} \, \|(a, b) - (a', b')\|_2.$$

Thus each $h_i$ is $\sqrt{2}$-Lipschitz with respect to the Euclidean norm.

Now apply Lemma A.5 with $K = 2$ and $L = \sqrt{2}$:

$$\widehat{\mathfrak{R}}_S(\mathcal{G}) = \frac{1}{N} \, \mathbb{E}_\sigma \left[ \sup_{u \in \mathcal{U}, \, v \in \mathcal{V}} \sum_{i=1}^{N} \sigma_i \psi(u(x_i), v(x_i)) \right] \leq \frac{2}{N} \, \mathbb{E}_{\sigma_{i1}, \sigma_{i2}} \left[ \sup_{u \in \mathcal{U}, \, v \in \mathcal{V}} \sum_{i=1}^{N} (\sigma_{i1} L_u u(x_i) + \sigma_{i2} L_v v(x_i)) \right].$$

Using $\sup(A + B) \leq \sup A + \sup B$, we obtain

$$\widehat{\mathfrak{R}}_S(\mathcal{G}) \leq \frac{2L_u}{N} \, \mathbb{E}_{\sigma_{i1}} \left[ \sup_{u \in \mathcal{U}} \sum_{i=1}^{N} \sigma_{i1} u(x_i) \right] + \frac{2L_v}{N} \, \mathbb{E}_{\sigma_{i2}} \left[ \sup_{v \in \mathcal{V}} \sum_{i=1}^{N} \sigma_{i2} v(x_i) \right].$$

Since $\{\sigma_{i1}\}_{i=1}^{N}$ and $\{\sigma_{i2}\}_{i=1}^{N}$ are again independent Rademacher families, this is exactly

$$\widehat{\mathfrak{R}}_S(\mathcal{G}) \leq 2L_u \widehat{\mathfrak{R}}_S(\mathcal{U}) + 2L_v \widehat{\mathfrak{R}}_S(\mathcal{V}).$$

Taking expectation over $S \sim p^N$ yields

$$\mathfrak{R}_N(\mathcal{G}, p) \leq 2L_u \mathfrak{R}_N(\mathcal{U}, p) + 2L_v \mathfrak{R}_N(\mathcal{V}, p).$$

$\square$

## A.1. Proof of Proposition 3.1

*Proof.* Make the change of variables $x_1 = x_0 + \sqrt{\varepsilon} \, z$, so that $dx_1 = \varepsilon^{D/2} dz$ and hence

$$Z_f(x_0) = (2\pi\varepsilon)^{D/2} \, \mathbb{E}_{z \sim \mathcal{N}(0, I)} \left[ \exp\left( \frac{f(x_0 + \sqrt{\varepsilon} z)}{\varepsilon} \right) \right] = \mathbb{E}_{z \sim \mathcal{N}(0, I)} [\exp(A(z))],$$

where

$$A(z) := \frac{f(x_0 + \sqrt{\varepsilon} z)}{\varepsilon} + \underbrace{\frac{D}{2} \log(2\pi\varepsilon)}_{\stackrel{\text{def}}{=} C}.$$

Then for any $\xi(x_0) \in \mathbb{R}$,

$$\log Z_f(x_0) = \log \mathbb{E}_{z \sim \mathcal{N}(0, I)}[e^A] = (\xi(x_0) + C) + \log \mathbb{E}[e^{A - (\xi(x_0) + C)}] \leq (\xi(x_0) + C) + \left( \mathbb{E}[e^{A - (\xi(x_0) + C)}] - 1 \right),$$

using pointwise $\log u \leq u - 1$ for all $u > 0$. Equality holds iff $\mathbb{E}[e^{A - (\xi + C)}] = 1$, i.e. when $\xi + C = \log \mathbb{E}[e^A] = \log Z_f(x_0)$. $\square$

## A.2. Proof of Theorem 3.2

*Proof.* Recall the weak dual form of entropic OT

$$\text{EOT}_\varepsilon(p_0, p_1) = \sup_f \left\{ \mathbb{E}_{x_1 \sim p_1}[f(x_1)] - \varepsilon \, \mathbb{E}_{x_0 \sim p_0} \left[ \log Z(f, x_0) \right] \right\},$$

where $Z(f, x_0) = \int_{\mathbb{R}^D} \exp\left( (f(x_1) - \frac{1}{2}\|x_1 - x_0\|^2)/\varepsilon \right) dx_1$. By Proposition 3.1 with $C = \frac{D}{2} \log(2\pi\varepsilon)$, for all $x_0 \in \mathbb{R}^D$,

$$\log Z(f, x_0) \leq C - 1 + \xi(x_0) + \mathbb{E}_{z \sim \mathcal{N}(0, I)} \left[ \exp\left( \frac{f(x_0 + \sqrt{\varepsilon} z)}{\varepsilon} - \xi(x_0) \right) \right].$$

Plugging this upper bound into the weak dual (and using that it holds pointwise in $x_0$) yields

$$\text{EOT}_\varepsilon(p_0, p_1) \geq \sup_{f,\xi} \mathcal{L}(f, \xi).$$

Conversely, since the inequality holds for every $\xi$, for each fixed $f$ we have $\mathcal{L}(f, \xi) \leq \mathbb{E}_{p_1}[f] - \varepsilon \mathbb{E}_{p_0}[\log Z(f, x_0)]$, hence $\sup_{f,\xi} \mathcal{L}(f, \xi) \leq \text{EOT}_\varepsilon(p_0, p_1)$. Finally, the bound is tight at $\xi(x_0) + C = \log Z(f, x_0)$ by Proposition 3.1, so equality holds. Defining $\mathcal{L}(f) := \sup_\xi \mathcal{L}(f, \xi)$ gives $\sup_f \mathcal{L}(f) = \sup_{f,\xi} \mathcal{L}(f, \xi)$. $\square$

### A.3. Proof of Theorem 3.3

*Proof.* Let the sup in the dual problem (2) be taken over a function $f^* \in L_1(p_1)$ (non-continuous in general), and set $\xi^*(x_0) := \xi_{f^*}$. Note that $\pi^* = \pi^{f^*} = \pi^{f^*, \xi^*}$. Therefore,

$$\varepsilon \int \log\left(\frac{d\pi^*}{d\pi^{f,\xi}}\right) d\pi^* = \varepsilon \int_{X_0} (\xi(x_0) - \xi^*(x_0))\, dp_0(x_0) + \int_{X_1} (f^*(x_1) - f(x_1))\, dp_1(x_1),$$

where we used that $\pi^*$ has marginals $p_0$ and $p_1$.

Next, we compute the total mass of $\pi^{f,\xi}$:

$$\int_{\mathbb{R}^D} \int_{\mathbb{R}^D} \frac{p_0(x)}{(2\pi\varepsilon)^{D/2}} \exp\left(\frac{f(x_1) - \frac{1}{2}\|x_0 - x_1\|^2}{\varepsilon} - \xi(x_0)\right) dx_1\, dx_0 = \int_{X_0} \frac{Z_f(x_0)}{(2\pi\varepsilon)^{D/2} \exp \xi(x_0)}\, dp_0(x_0).$$

Since $\pi^*$ is a probability measure. Combining the pieces, we obtain

$$\varepsilon \text{KL}(\pi^* \| \pi^{f,\xi}) = \left(\int_{\mathbb{R}^D} f^*\, dp_1 - \varepsilon \int_{\mathbb{R}^D} \xi^*\, dp_0\right) - \left(\int_{\mathbb{R}^D} f\, dp_1 - \varepsilon \int_{\mathbb{R}^D} \xi\, dp_0 - \varepsilon \int_{\mathbb{R}^D} \frac{Z_f}{(2\pi\varepsilon)^{D/2} \exp \xi}\, dp_0 + \varepsilon\right).$$

Finally, since $(2\pi\varepsilon)^{D/2} \exp \xi^* = Z_{f^*}$, the first bracket equals $\mathcal{L}^* + \frac{\varepsilon D}{2} \log(2\pi\varepsilon)$, while the second bracket is precisely $\mathcal{L}(f, \xi) + \frac{\varepsilon D}{2} \log(2\pi\varepsilon)$ as defined in (12). Hence,

$$\varepsilon \text{KL}\left(\pi^* \,\big\|\, \pi^{f,\xi}\right) = \mathcal{L}^* - \mathcal{L}(f, \xi),$$

which proves (14). $\square$

For the reader's convenience, we briefly recall the empirical loss

$$\widehat{\mathcal{L}}(f, \xi) = \varepsilon - \frac{\varepsilon D}{2} \log(2\pi\varepsilon) + \frac{1}{N_1} \sum_{j=1}^{N_1} f(x_j^1) - \frac{\varepsilon}{N_0} \sum_{i=1}^{N_0} \xi(x_i^0) - \frac{\varepsilon}{N_0 K} \sum_{i,k=1}^{N_0, K} \exp\left(\frac{f(x_i^0 + \sqrt{\varepsilon}\, z_{i,k})}{\varepsilon} - \xi(x_i^0)\right), \quad (20)$$

where the samples are taken from the corresponding distributions $x_j^1 \sim p_1$, $x_i^0 \sim p_0$, $z_{i,k} \sim \mathcal{N}(0, I)$.

### A.4. Proof of Proposition 3.4

*Proof.* We assume that all weight and bias parameters belong to closed and bounded sets, hence the parameter sets of $\mathcal{F}$ and $\Xi$ are compact. Let

$$(\hat{f}, \hat{\xi}) = \arg\max_{f \in \mathcal{F}, \xi \in \Xi} \widehat{\mathcal{L}}(f, \xi),$$

and recall the notation $\xi_f = \log Z_f - \frac{D}{2} \log(2\pi\varepsilon)$. Since $\xi_f$ maximizes $\mathcal{L}(f, \xi)$ over $\xi$ for fixed $f$, we have

$$\mathcal{L}(\hat{f}, \xi_{\hat{f}}) \geq \mathcal{L}(\hat{f}, \hat{\xi}).$$

Therefore, by Theorem 3.3

$$\varepsilon \text{KL}\left(\pi^* \big\| \pi^{\hat{f}}\right) = \mathcal{L}^* - \mathcal{L}(\hat{f}, \xi_{\hat{f}}) \leq \mathcal{L}^* - \mathcal{L}(\hat{f}, \hat{\xi}) = \mathcal{L}^* - \sup_{f \in \mathcal{F},\, \xi \in \Xi} \mathcal{L}(f, \xi) + \underbrace{\sup_{f \in \mathcal{F},\, \xi \in \Xi} \mathcal{L}(f, \xi) - \mathcal{L}(\hat{f}, \hat{\xi})}_{(*)}.$$

$$(*) = \sup_{f \in \mathcal{F}, \xi \in \Xi} \mathcal{L}(f, \xi) - \sup_{f \in \mathcal{F}, \xi \in \Xi} \widehat{\mathcal{L}}(f, \xi) + \widehat{\mathcal{L}}(\hat{f}, \hat{\xi}) - \mathcal{L}(\hat{f}, \hat{\xi}) \leq \sup_{f \in \mathcal{F}, \xi \in \Xi} \left[ \mathcal{L}(f, \xi) - \widehat{\mathcal{L}}(f, \xi) \right] + \underbrace{\widehat{\mathcal{L}}(\hat{f}, \hat{\xi}) - \mathcal{L}(\hat{f}, \hat{\xi})}_{\leq \sup_{f \in \mathcal{F}, \xi \in \Xi} |\mathcal{L}(f,\xi) - \widehat{\mathcal{L}}(f,\xi)|}$$

Combining the above estimates yields

$$\varepsilon \operatorname{KL}\left(\pi^* \big\| \pi^{\hat{f}}\right) \leq \mathcal{L}^* - \sup_{f \in \mathcal{F}, \ \xi \in \Xi} \mathcal{L}(f, \xi) + 2 \sup_{f \in \mathcal{F}, \ \xi \in \Xi} \left| \mathcal{L}(f, \xi) - \widehat{\mathcal{L}}(f, \xi) \right|.$$

Taking expectation completes the proof. $\qquad\square$

### A.5. Proof of Theorem 3.5

*Proof.* Let $M_{\mathcal{F}}$ and $M_{\Xi}$ be clipping constants in classes $\mathcal{F}$ and $\Xi$ respectively.

Set

$$h_{f,\xi}(x, z) := \exp\left( \frac{f(x + \sqrt{\varepsilon} z)}{\varepsilon} - \xi(x) \right), \qquad g_{f,\xi}(x) := \mathbb{E}_{z \sim \mathcal{N}(0,I)} h_{f,\xi}(x, z).$$

Then

$$\mathcal{L}(f, \xi) - \widehat{\mathcal{L}}(f, \xi) = \left( \mathbb{E}_{p_1} f - \frac{1}{N_1} \sum_{j=1}^{N_1} f(x_j^1) \right) - \varepsilon \left( \mathbb{E}_{p_0} \xi - \frac{1}{N_0} \sum_{i=1}^{N_0} \xi(x_i^0) \right) - \varepsilon \left( \mathbb{E}_{p_0} g_{f,\xi} - \frac{1}{N_0 K} \sum_{i,k} h_{f,\xi}(x_i^0, z_{ik}) \right).$$

Hence, by the triangle inequality,

$$\sup_{f \in \mathcal{F}, \ \xi \in \Xi} \left| \mathcal{L}(f, \xi) - \widehat{\mathcal{L}}(f, \xi) \right| \leq \Delta_1 + \varepsilon \Delta_2 + \varepsilon \Delta_3 + \varepsilon \Delta_4,$$

where

$$\Delta_1 := \sup_{f \in \mathcal{F}} \left| \mathbb{E}_{p_1} f - \frac{1}{N_1} \sum_{j=1}^{N_1} f(x_j^1) \right|, \quad \Delta_2 := \sup_{\xi \in \Xi} \left| \mathbb{E}_{p_0} \xi - \frac{1}{N_0} \sum_{i=1}^{N_0} \xi(x_i^0) \right|,$$

$$\Delta_3 := \sup_{f,\xi} \left| \mathbb{E}_{p_0} g_{f,\xi} - \frac{1}{N_0} \sum_{i=1}^{N_0} g_{f,\xi}(x_i^0) \right|, \quad \Delta_4 := \sup_{f,\xi} \left| \frac{1}{N_0} \sum_{i=1}^{N_0} g_{f,\xi}(x_i^0) - \frac{1}{N_0 K} \sum_{i=1}^{N_0} \sum_{k=1}^{K} h_{f,\xi}(x_i^0, z_{ik}) \right|.$$

By Lemma A.4,

$$\mathbb{E}[\Delta_1] \leq 2 \, \mathfrak{R}_{N_1}(\mathcal{F}, p_1), \quad \mathbb{E}[\Delta_2] \leq 2 \, \mathfrak{R}_{N_0}(\Xi, p_0),$$

where all bounds are taken in expectation with respect to the random samples.

Recall that the classes $\mathcal{F}$ and $\Xi$ consist of feedforward neural networks with depth at most $n$, with uniformly bounded layer norms, and with output clipping. Also the hidden activation $\sigma$ be $L_\sigma$-Lipschitz and satisfies $\sigma(0) = 0$, and that the layer matrices satisfy the row-wise constraints from (Golowich et al., 2018, Theorem 2). Then there exists a constants $C_{\mathcal{F}}^{p_1} > 0$ and $C_{\Xi}^{p_0} > 0$ such that

$$\mathbb{E}[\Delta_1] \leq \frac{C_{\mathcal{F}}^{p_1}}{\sqrt{N_1}}, \quad \mathbb{E}[\Delta_2] \leq \frac{C_{\Xi}^{p_0}}{\sqrt{N_0}}. \tag{21}$$

**Estimation of $\Delta_3$.** Let $\gamma_\varepsilon = \mathcal{N}(0, \varepsilon I)$ and let $\rho_\varepsilon(y)$ be its density. Consider the classes (note that $\gamma_1 = \mathcal{N}(0, I)$)

$$\mathcal{Z} := \left\{ z_f := \int \exp\left( \frac{f(x + \sqrt{\varepsilon} z)}{\varepsilon} \right) d\gamma_1(z) : f \in \mathcal{F} \right\}, \quad \mathcal{E} := \{ e^{-\xi} : \xi \in \Xi \}, \quad \mathcal{G} := \{ g_{f,\xi} = e^{-\xi} z_f : f \in \mathcal{F}, \ \xi \in \Xi \}.$$

By Lemma A.4,

$$\mathbb{E}[\Delta_3] \leq 2 \, \mathfrak{R}_{N_0}(\mathcal{G}, p_0).$$

For $x \in X_0$, define $q_x(y) := \rho_\varepsilon(y - x) / \rho_\varepsilon(y)$ and for $f \in \mathcal{F}$, define

$$\phi_f(y) := \exp\left( \frac{f(y)}{\varepsilon} \right).$$

Then

$$z_f(x) = \int_{\mathbb{R}^D} \phi_f(y)\, \rho_\varepsilon(y - x)\, dy = \int_{\mathbb{R}^D} \phi_f(y)\, q_x(y)\, d\gamma_\varepsilon(y) = \langle \phi_f, q_x \rangle_{L^2(\gamma_\varepsilon)}.$$

Since $|f| \le M_{\mathcal{F}}$, we have

$$\|\phi_f\|_{L^2(\gamma_\varepsilon)} \le e^{M_{\mathcal{F}}/\varepsilon}, \qquad \|q_x\|^2_{L^2(\gamma_\varepsilon)} = \exp\left(\frac{\|x\|^2}{\varepsilon}\right)$$

Therefore,

$$\sup_{x \in \mathrm{supp}(p_0)} \|q_x\|_{L^2(\gamma_\varepsilon)} \le \exp\left(\frac{R_0^2}{2\varepsilon}\right), \quad \text{where } R_0 = \max_{x \in \mathrm{supp}(p_0)} \|x\|$$

Using the representation above and given $\{\sigma_i\}_{i=1}^{N_0}$ – i.i.d. Rademacher signs

$$\frac{1}{N_0} \sum_{i=1}^{N_0} \sigma_i z_f(x_i) = \left\langle \phi_f, \frac{1}{N_0} \sum_{i=1}^{N_0} \sigma_i q_{x_i} \right\rangle_{L^2(\gamma_\varepsilon)}.$$

Hence, by Cauchy–Schwarz,

$$\mathfrak{R}_{N_0}(\mathcal{Z}, p_0) \le e^{M_{\mathcal{F}}/\varepsilon}\, \mathbb{E}_{x,\sigma}\left[\left\|\frac{1}{N_0}\sum_{i=1}^{N_0}\sigma_i q_{x_i}\right\|_{L^2(\gamma_\varepsilon)}\right].$$

Applying Jensen's inequality, the independence of $\sigma_i$ and $\mathbb{E}\sigma_i = 0$ of Rademacher signs, we get

$$\mathfrak{R}_{N_0}(\mathcal{Z}, p_0) \le e^{M_{\mathcal{F}}/\varepsilon}\left(\mathbb{E}\left\|\frac{1}{N_0}\sum_{i=1}^{N_0}\sigma_i q_{x_i}\right\|^2_{L^2(\gamma_\varepsilon)}\right)^{1/2} = e^{M_{\mathcal{F}}/\varepsilon}\left(\frac{1}{N_0^2}\sum_{i=1}^{N_0}\mathbb{E}\|q_{x_i}\|^2_{L^2(\gamma_\varepsilon)}\right)^{1/2} \le \frac{e^{M_{\mathcal{F}}/\varepsilon + R_0^2/(2\varepsilon)}}{\sqrt{N_0}}.$$

Note that $|\xi| \le M_\Xi$, the function $\phi(t) := e^{-t}$ is $e^{M_\Xi}$-Lipschitz on $[-M_\Xi, M_\Xi]$, and $\mathfrak{R}_{N_0}(\mathcal{E}, p_0) = \mathfrak{R}_{N_0}(\phi \circ \Xi, p_0)$. Hence, by Lemma A.5 and (21),

$$\mathfrak{R}_{N_0}(\mathcal{E}, p_0) \le \sqrt{2}\, e^{M_\Xi}\, \mathfrak{R}_{N_0}(\Xi, p_0) \le \frac{\sqrt{2}\, e^{M_\Xi} C_\Xi^{p_0}}{\sqrt{N_0}}.$$

Note that the classes $\mathcal{E}$ and $\mathcal{Z}$ are uniformly bounded

$$e^{-M_\Xi} \le e^{-\xi(x)} \le e^{M_\Xi}, \qquad 0 \le z_f(x) \le e^{M_{\mathcal{F}}/\varepsilon}.$$

Consider the map $\psi(u, v) := uv$ on the rectangle $[0, e^{M_{\mathcal{F}}/\varepsilon}] \times [e^{-M_\Xi}, e^{M_\Xi}]$. It is coordinatewise Lipschitz with constants $L_u = e^{M_\Xi}$ and $L_v = e^{M_{\mathcal{F}}/\varepsilon}$. Applying Lemma A.6 to

$$\mathcal{Z} \times \mathcal{E} = \{(z_f, e^{-\xi}) : f \in \mathcal{F}, \xi \in \Xi\},$$

we obtain

$$\mathfrak{R}_{N_0}(\mathcal{G}, p_0) \le 2e^{M_\Xi}\, \mathfrak{R}_{N_0}(\mathcal{Z}, p_0) + 2e^{M_{\mathcal{F}}/\varepsilon}\, \mathfrak{R}_{N_0}(\mathcal{E}, p_0).$$

Combining this with the bounds above yields

$$\mathfrak{R}_{N_0}(\mathcal{G}, p_0) \le \frac{2e^{M_{\mathcal{F}}/\varepsilon + M_\Xi}}{\sqrt{N_0}}\left(e^{R_0^2/2\varepsilon} + \sqrt{2}\, C_\Xi^{p_0}\right).$$

**Estimation of $\Delta_4$.** Fix $x \in X_0$ and set

$$\mathcal{F}_x := \{z \mapsto f(x + \sqrt{\varepsilon}z) : f \in \mathcal{F}\}.$$

Since $\xi(x)$ does not depend on $z$ and $|\xi| \le M_\Xi$, we have

$$\sup_{f,\xi}\left|\mathbb{E}_z h_{f,\xi}(x, z) - \frac{1}{K}\sum_{k=1}^K h_{f,\xi}(x, z_k)\right| \le e^{M_\Xi}\sup_{f \in \mathcal{F}}\left|\mathbb{E}_z e^{f(x+\sqrt{\varepsilon}z)/\varepsilon} - \frac{1}{K}\sum_{k=1}^K e^{f(x+\sqrt{\varepsilon}z_k)/\varepsilon}\right|.$$

By symmetrization and the contraction inequality applied to $t \mapsto e^{t/\varepsilon}$ on $[-M_{\mathcal{F}}, M_{\mathcal{F}}]$, whose Lipschitz constant is $e^{M_{\mathcal{F}}/\varepsilon}/\varepsilon$, we get

$$\mathbb{E}_z \sup_{f,\xi} \left| \mathbb{E}_z h_{f,\xi}(x,z) - \frac{1}{K} \sum_{k=1}^{K} h_{f,\xi}(x, z_k) \right| \leq \frac{C e^{M_{\Xi}+M_{\mathcal{F}}/\varepsilon}}{\varepsilon} \mathfrak{R}_K(\mathcal{F}_x, \gamma_1),$$

where $\gamma_1 = \mathcal{N}(0, I)$ and $C > 0$ is a universal constant.

We now use only the sample-dependent Rademacher bound from (Golowich et al., 2018, Theorem 1). In the notation adapted to our network class, it gives a constant $A_{\mathcal{F}} > 0$, depending only on the depth and the layer-norm bounds of $\mathcal{F}$, such that for every deterministic sample $y_1, \ldots, y_K \in \mathbb{R}^D$,

$$\widehat{\mathfrak{R}}_{\{y_k\}_{k=1}^{K}}(\mathcal{F}) \leq \frac{A_{\mathcal{F}}}{K} \left( \sum_{k=1}^{K} \|y_k\|^2 \right)^{1/2}.$$

Applying this inequality to $y_k = x + \sqrt{\varepsilon} z_k$ yields

$$\mathfrak{R}_K(\mathcal{F}_x, \gamma_1) \leq \frac{A_{\mathcal{F}}}{K} \mathbb{E} \left( \sum_{k=1}^{K} \|x + \sqrt{\varepsilon} Z_k\|^2 \right)^{1/2}.$$

By Jensen's inequality,

$$\mathbb{E} \left( \sum_{k=1}^{K} \|x + \sqrt{\varepsilon} Z_k\|^2 \right)^{1/2} \leq \left( K \mathbb{E} \|x + \sqrt{\varepsilon} Z\|^2 \right)^{1/2} = \sqrt{K} \left( \|x\|^2 + \varepsilon D \right)^{1/2}.$$

Since $X_0$ is compact, $R_0 := \sup_{x \in X_0} \|x\| < \infty$, and therefore

$$\sup_{x \in X_0} \mathfrak{R}_K(\mathcal{F}_x, \gamma) \leq \frac{A_{\mathcal{F}} (R_0^2 + \varepsilon D)^{1/2}}{\sqrt{K}}.$$

Consequently,

$$\mathbb{E}[\Delta_4] \leq \frac{C e^{M_{\Xi}+M_{\mathcal{F}}/\varepsilon}}{\varepsilon} \frac{A_{\mathcal{F}} (R_0^2 + \varepsilon D)^{1/2}}{\sqrt{K}}.$$

Combining the bounds for $\Delta_1, \Delta_2, \Delta_3, \Delta_4$, we obtain constants $C_1, C_2, C_3 > 0$, depending only on $\varepsilon$, the clipping levels, the network-complexity parameters, and the support bounds, such that

$$\mathbb{E} \sup_{f \in \mathcal{F}, \xi \in \Xi} \left| \mathcal{L}(f, \xi) - \widehat{\mathcal{L}}(f, \xi) \right| \leq \frac{C_1}{\sqrt{N_1}} + \frac{C_2}{\sqrt{N_0}} + \frac{C_3}{\sqrt{K}}.$$

This proves the claimed $O(N_1^{-1/2}) + O(N_0^{-1/2}) + O(K^{-1/2})$ bound.

$\square$

### A.6. Proof of Theorem 3.6

*Proof.* It is sufficient to take the supremum in the weak dual problem (2) over continuous potentials. Indeed, under the standing admissibility assumptions, every admissible measurable potential can first be approximated, in the value of the functional, by its bounded truncations. For bounded potentials, approximation by continuous functions in the weighted space naturally associated with the Gaussian kernel in $Z_f$ preserves both terms of

$$\mathcal{L}(f) = \int f(x_1) \, dp_1(x_1) - \varepsilon \int \log Z_f(x_0) \, dp_0(x_0).$$

The first term is stable by $L^1(p_1)$-convergence, and the log-partition term is stable because the exponential is uniformly Lipschitz on bounded ranges. Hence the value of the supremum is unchanged if the admissible measurable class is replaced by $C(\mathbb{R}^D)$.

Fix $\delta > 0$. Then there exist $f \in C(\mathbb{R}^D)$ such that

$$\mathcal{L}^* - \mathcal{L}(f, \xi_f) < \frac{\delta}{3}.$$

Recall that

$$\xi_f(x_0) = \log \mathbb{E}_{z \sim \mathcal{N}(0,I)} \exp\left(\frac{f(x_0 + \sqrt{\varepsilon} z)}{\varepsilon}\right),$$

so $\xi_f$ is continuous in our setting. Further in the text, we will omit the index at $\xi_f$ and set $\xi := \xi_f$. Denote

$$\Phi(f, \xi) := \mathbb{E}_{x_0 \sim p_0} \mathbb{E}_{z \sim \mathcal{N}(0,I)} \exp\left(\frac{f(x_0 + \sqrt{\varepsilon} z)}{\varepsilon} - \xi(x_0)\right).$$

Then

$$\mathcal{L}(f, \xi) = \varepsilon\left(1 - \frac{D}{2}\log(2\pi\varepsilon)\right) + \mathbb{E}_{p_1}[f] - \varepsilon \mathbb{E}_{p_0}[\xi] - \varepsilon\, \Phi(f, \xi).$$

Since $f$ and $\xi$ are continuous and $X_1, X_0$ are compact, the restrictions $f|_{X_1}$ and $\xi|_{X_0}$ are bounded. Choose $M > 0$ so that

$$M > \|f\|_{L^\infty(X_1)} + \|\xi\|_{L^\infty(X_0)} \qquad \text{and} \qquad \varepsilon e^{\|\xi\|_{L^\infty(X_0)}} e^{-M/\varepsilon} < \frac{\delta}{6}.$$

Let

$$T_M(t) := (-M) \vee t \wedge M, \qquad \bar{f} := T_M(f).$$

Then $\bar{f} \in C(\mathbb{R}^D)$ and $|\bar{f}| \leq M$ on $\mathbb{R}^D$. Moreover, $\bar{f} = f$ on $X_1$, hence the linear term $\mathbb{E}_{p_1}[f]$ does not change. Also, clipping from above can only decrease the exponential term in $\Phi$, while clipping from below changes it pointwise by at most $e^{\|\xi\|_{L^\infty(X_0)}} e^{-M/\varepsilon}$. Therefore

$$\Phi(\bar{f}, \xi) \leq \Phi(f, \xi) + e^{\|\xi\|_{L^\infty(X_0)}} e^{-M/\varepsilon},$$

and so

$$\mathcal{L}(\bar{f}, \xi) \geq \mathcal{L}(f, \xi) - \varepsilon e^{\|\xi\|_{L^\infty(X_0)}} e^{-M/\varepsilon} > \mathcal{L}^* - \frac{\delta}{2}.$$

Next define the probability measure

$$q(dx_1) := \left(\int_{X_0} (2\pi\varepsilon)^{-D/2} \exp\left(-\frac{\|x_1 - x_0\|^2}{2\varepsilon}\right) p_0(dx_0)\right) dx_1.$$

Let

$$L_M := e^{M/\varepsilon + M} \max\{\varepsilon^{-1}, 1\}.$$

If $|f_1|, |f_2| \leq M$ on $\mathbb{R}^D$ and $|\xi_1|, |\xi_2| \leq M$ on $X_0$, then

$$\left|\Phi(f_1, \xi_1) - \Phi(f_2, \xi_2)\right| \leq L_M \left(\int_{\mathbb{R}^D} |f_1 - f_2|\, dq + \int_{X_0} |\xi_1 - \xi_2|\, dp_0\right).$$

Consequently,

$$\begin{aligned}
|\mathcal{L}(f_1, \xi_1) - \mathcal{L}(f_2, \xi_2)| \leq{}& \int_{X_1} |f_1 - f_2|\, dp_1 + \varepsilon \int_{X_0} |\xi_1 - \xi_2|\, dp_0 \\
&+ \varepsilon L_M \left(\int_{\mathbb{R}^D} |f_1 - f_2|\, dq + \int_{X_0} |\xi_1 - \xi_2|\, dp_0\right).
\end{aligned} \tag{22}$$

Choose $\alpha > 0$ and then $R > 0$ so large that

$$X_1 \subset B_R \qquad \text{and} \qquad q(\mathbb{R}^d \setminus B_R) < \alpha.$$

Since $\sigma$ is Lipschitz, it is continuous; since it is also non-polynomial, the universal approximation theorem for one-hidden-layer ridge networks with biases applies, see (Pinkus, 1999, Theorem 3.1). More precisely, the theorem concerns the family

$$M(\sigma) = \text{span}\{x \mapsto \sigma(w^\top x + b): w \in \mathbb{R}^D, b \in \mathbb{R}\},$$

which is dense in $C(K)$ for every compact $K \subset \mathbb{R}^D$.

Hence there exist one-hidden-layer networks

$$g_\theta(x) = \sum_{k=1}^{m_1} a_k \, \sigma(w_k^\top x + b_k), \qquad h_\vartheta(x) = \sum_{\ell=1}^{m_2} c_\ell \, \sigma(u_\ell^\top x + d_\ell),$$

such that

$$\sup_{x \in B_R} |g_\theta(x) - \bar{f}(x)| < \alpha, \qquad \sup_{x \in X_0} |h_\vartheta(x) - \xi(x)| < \alpha.$$

Now perform clipping after approximation and define

$$f_\theta := T_M \circ g_\theta, \qquad \xi_\vartheta := T_M \circ h_\vartheta.$$

Since $|\bar{f}| \leq M$ on $\mathbb{R}^D$ and $|\xi| \leq M$ on $X_0$, clipping does not increase the approximation error on the target sets; thus

$$\sup_{x \in B_R} |f_\theta(x) - \bar{f}(x)| < \alpha, \qquad \sup_{x \in X_0} |\xi_\vartheta(x) - \xi(x)| < \alpha,$$

and, by construction,

$$|f_\theta(x)| \leq M \quad \forall x \in \mathbb{R}^D.$$

Therefore,

$$\int_{X_1} |f_\theta - \bar{f}| \, dp_1 \leq \alpha, \qquad \int_{X_0} |\xi_\vartheta - \xi| \, dp_0 \leq \alpha,$$

and

$$\int_{\mathbb{R}^D} |f_\theta - \bar{f}| \, dq \leq \alpha \, q(B_R) + 2M \, q(\mathbb{R}^d \setminus B_R) \leq \alpha + 2M\alpha.$$

Substituting these bounds into (22), we obtain

$$|\mathcal{L}(f_\theta, \xi_\vartheta) - \mathcal{L}(\bar{f}, \xi)| \leq \alpha + \varepsilon\alpha + \varepsilon L_M (2\alpha + 2M\alpha).$$

Choosing $\alpha > 0$ sufficiently small, we get

$$|\mathcal{L}(f_\theta, \xi_\vartheta) - \mathcal{L}(\bar{f}, \xi)| < \frac{\delta}{2}.$$

Hence

$$\mathcal{L}(f_\theta, \xi_\vartheta) > \mathcal{L}^* - \delta.$$

Finally, choose neural-network classes $\mathcal{F}$ and $\Xi$ so that they contain the clipped networks $f_\theta$ and $\xi_\vartheta$ constructed above. Then

$$\sup_{f \in \mathcal{F}, \, \xi \in \Xi} \mathcal{L}(f, \xi) \geq \mathcal{L}(f_\theta, \xi_\vartheta) > \mathcal{L}^* - \delta.$$

This proves the claim.

$$\square$$

## B. Details of the Experiments

### B.1. VarEOT: Optimization and Architecture

**Optimization.** Training is performed using the AdamW (Loshchilov & Hutter, 2017) optimizer with a learning rate $\text{lr} = 10^{-4}$, momentum parameters $\beta_1 = 0.7$ and $\beta_2 = 0.8$, and weight decay set to $10^{-4}$. We additionally employ an exponential moving average (EMA) of model parameters with momentum $\text{ema\_momentum} = 0.999$, which is used for evaluation. All models are trained for $10^4$ gradient steps.

**Network architecture.** The transport potential is parameterized by a multilayer perceptron (MLP). The network consists of four fully connected layers with SiLU activations. The input dimensionality is denoted by $d_{\text{in}}$ (512 for latent-space experiments), the hidden layer width is fixed to 256 units, and the output dimension is $d_{\text{out}} = 1$.

**Training setup.**    All models are trained with a batch size of 256. The number of Monte Carlo samples used to estimate expectations is fixed to $K = 256$ in the image experiments, for MSCI we use $K = 128$.

**Langevin inference.**    The number of Langevin steps $S$ (denoted as NFE) and the corresponding step size are chosen depending on the experimental setting. For all toy experiments on the SwissRoll dataset, we use 1000 Langevin steps with a fixed step size of $10^{-3}$. For image-based experiments in the ALAE latent space, we consider multiple inference budgets. The step sizes for each NFE configuration are selected based on FID performance, as illustrated in Figure 7. Specifically, we use a step size of 0.5 for NFE $= 2$, a step size of 0.1 for NFE $= 10$, and a step size of $10^{-3}$ for NFE $= 1000$.

**Update schedule and practical considerations.**    We perform simultaneous optimization of both the dual potential network $\hat{f}_\theta$ and the auxiliary network $\hat{\xi}_\psi$ at every training step, which we found to lead to stable training and strong empirical performance. To further stabilize training, we clip the difference values appearing in the exponential term of (15) from above by 30, and apply standard gradient clipping with norm threshold 1.

### B.2. Baseline Methods

For completeness, we summarize the main technical details of the baseline methods used for comparison in the unpaired image-to-image translation experiments.

**LightSB.**    For LightSB, we follow the standard configuration reported in prior work. The model is trained using $K = 10$ Gaussian components, a learning rate of lr $= 10^{-3}$, and a batch size of 128. Optimization is performed for $10^4$ gradient steps.

**EgNOT.**    For EgNOT, we follow the optimization/hyperparameter setup from the original work, but adopt the same neural network architecture for potential $f$ as in section B.1 for fair comparison with VarEOT. Training/inference hyperparameters were selected by grid search over a reasonable parameter space based on FID performance. For all ALAE experiments, Adam optimizer is used with lr $= 5 \cdot 10^{-5}$, $(\beta_1, \beta_2) = (0, 0.999)$. Optimization is performed for $10^4$ gradient steps, batch size is 128.

### B.3. Image Data and Preprocessing

For image-based experiments, we rely on the official implementation of ALAE and the corresponding pretrained models, available at

https://github.com/podgorskiy/ALAE

To obtain semantic attributes for the FFHQ dataset, we use the neural network–based annotations released at

https://github.com/DCGM/ffhq-features-dataset

For baseline comparisons, we use the official implementations of LightSB and EgNOT, available at

https://github.com/ngushchin/LightSB

and

https://github.com/PetrMokrov/Energy-guided-Entropic-OT

respectively.

## C. Additional Experimental Results

### C.1. Wall-Clock Training Time Comparison

We report wall-clock training times for the M→F experiment ($\varepsilon{=}1.0$) with 10000 iterations, using the same hyperparameters as in Appendix B.1. For VarEOT, we vary the number of particles $K \in \{32, 64, 128, 256, 512\}$ to study the speed–quality trade-off. All experiments for VarEOT and EgNOT were conducted on a single GPU, while LightSB experiments were run on CPU, as in the original paper.

| Method | Config | Time | Iter | FID ↓ |
|---|---|---|---|---|
| VarEOT | $K{=}512$ | 409 s | 10000 | **9.575** |
| VarEOT | $K{=}256$ | 215 s | 10000 | **10.55** |
| VarEOT | $K{=}128$ | 131 s | 10000 | **11.43** |
| VarEOT | $K{=}64$ | 85 s | 10000 | **12.85** |
| VarEOT | $K{=}32$ | 85 s | 10000 | **13.45** |
| LightSB (Korotin et al., 2024) | — | 191 s | 10000 | 22.63 |
| EgNOT (Mokrov et al., 2024) | — | 1692 s | 10000 | 17.58 |

*Table 4.* Wall-clock training time and FID on the M→F task ($\varepsilon{=}1.0$; 10000 iterations). VarEOT training time scales with the number of particles $K$; larger $K$ yields better FID at the cost of longer training.

### C.2. Other Unpaired Image-to-Image Translation Setups

In the main text, we present qualitative results for the unpaired image-to-image translation task using the ALAE encoder in the Male→Female (M→F) setup. In this appendix, we provide additional qualitative examples for the same task under alternative and commonly used translation directions.

Specifically, we report results for the Woman→Man (F→M), Adult→Child (A→C), and Child→Adult (C→A) setups. All experiments are conducted under the same training protocol and model configuration as in the main text, and differ only in the choice of the source and target domains. For consistency, we fix the regularization parameter to $\varepsilon = 1.0$ for all setups shown below.

Figures 3–5 demonstrate that the proposed method successfully learns meaningful transport maps across different translation directions, further confirming that the qualitative behavior observed in the M→F setup generalizes to other unpaired image-to-image translation scenarios.

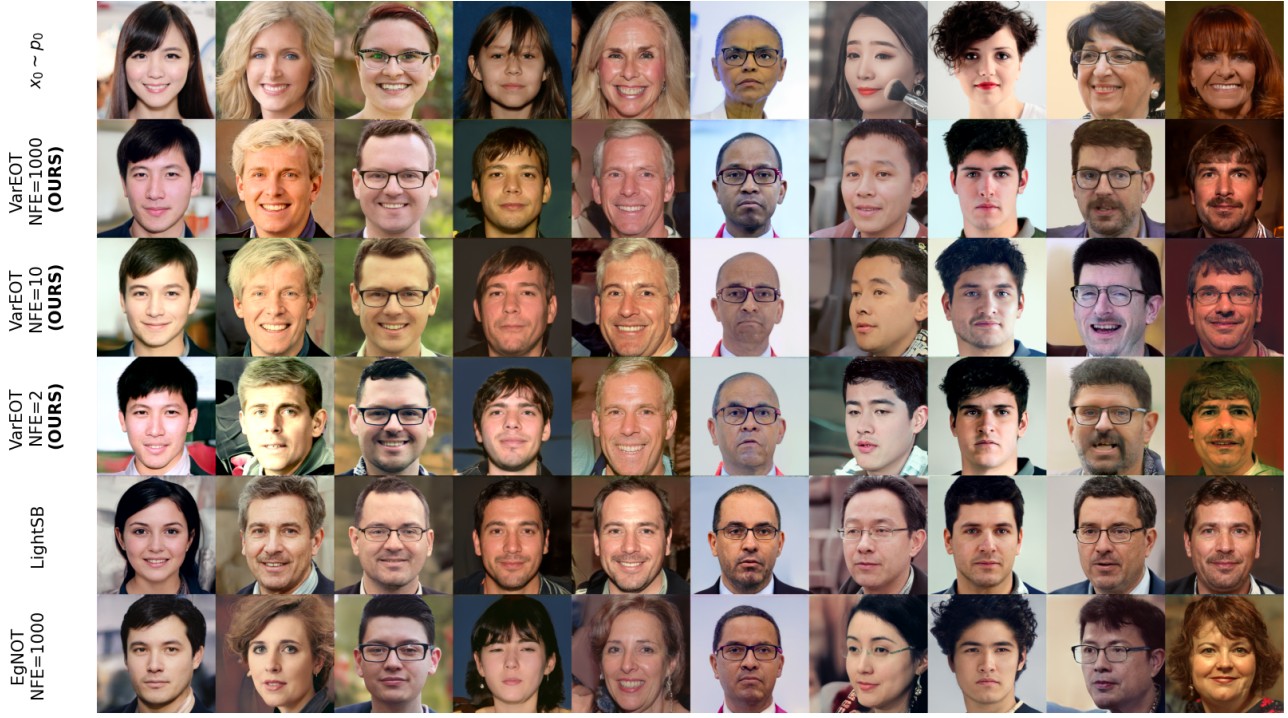

*Figure 3.* Qualitative comparison for *Female → Male* translation with $\varepsilon = 1.0$. From top to bottom: input samples, VarEOT (ours), LightSB, and EgNOT. Input images are selected from the test set: we take the first 300 samples and rank them by encoder-decoder reconstruction quality (LPIPS), displaying the top-ranked examples.

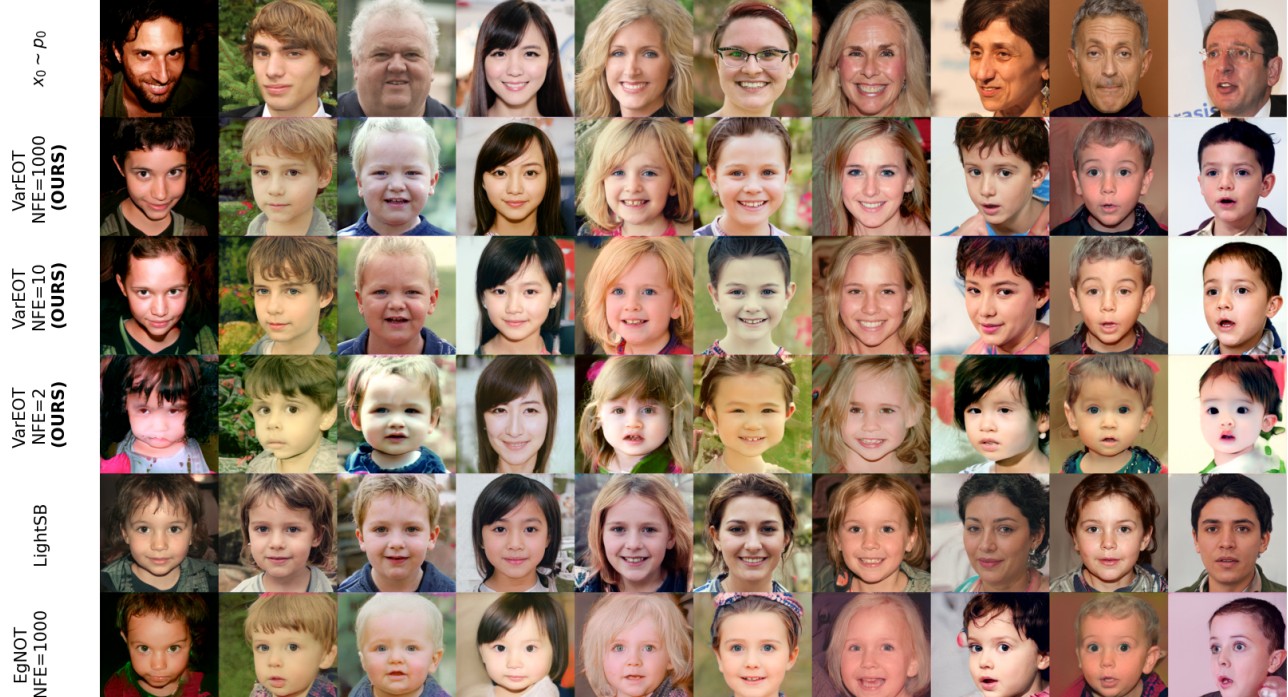

*Figure 4.* Qualitative comparison for *Adult → Child* translation with $\varepsilon = 1.0$. From top to bottom: input samples, VarEOT (ours), LightSB, and EgNOT. Input images are selected from the test set: we take the first 300 samples and rank them by encoder-decoder reconstruction quality (LPIPS), displaying the top-ranked examples.

## C.3. Dependence on the parameter $\varepsilon$.

In Figure 6, we show how the solution learned by VarEOT depends on the parameter $\varepsilon$ in the *Male→Female* experiment. As expected, the diversity increases with the increase of $\varepsilon$.

## C.4. Effect of Langevin Inference Parameters

To further analyze the behavior of VarEOT at inference time, we study the sensitivity of the translation quality to the Langevin sampling parameters in the *Male→Female* (M→F) image-to-image translation task. Figure 7 reports heatmaps of FID scores as a function of the Langevin step size and the number of sampling steps, for different values of the entropic regularization parameter $\varepsilon$.

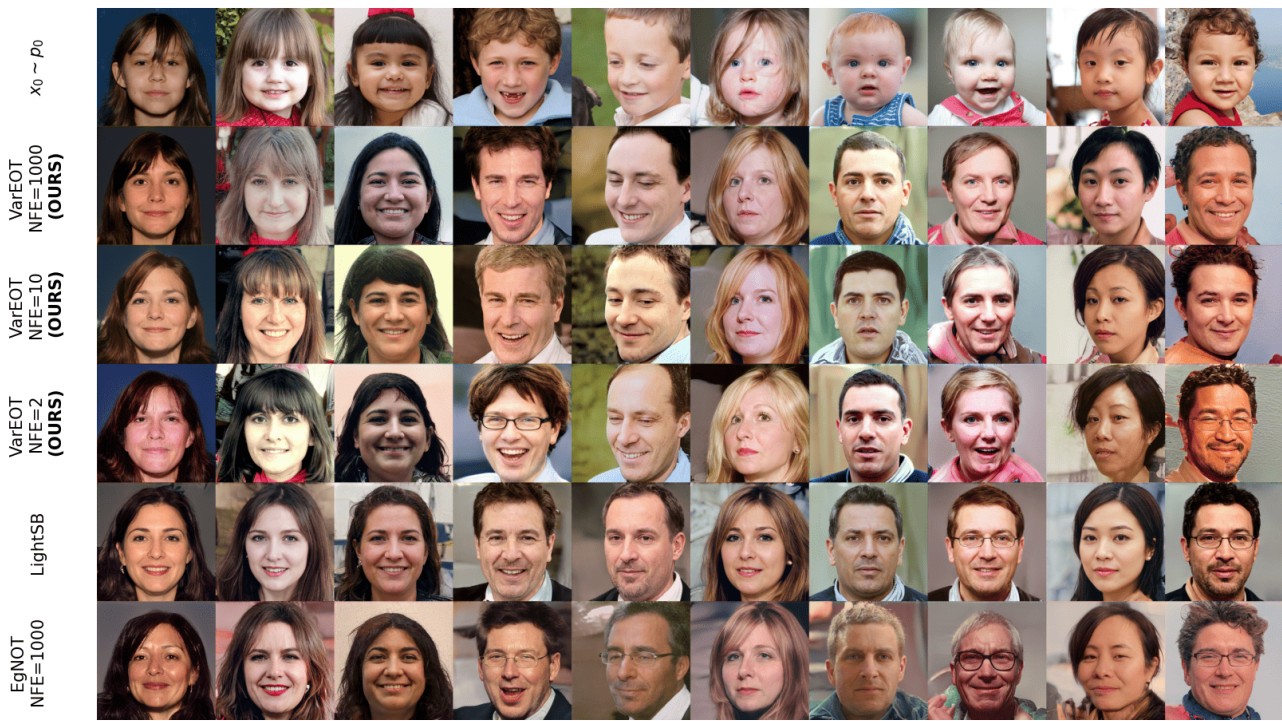

*Figure 5.* Qualitative comparison for *Child → Adult* translation with $\varepsilon = 1.0$. From top to bottom: input samples, VarEOT (ours), LightSB, and EgNOT. Input images are selected from the test set: we take the first 300 samples and rank them by encoder-decoder reconstruction quality (LPIPS), displaying the top-ranked examples.

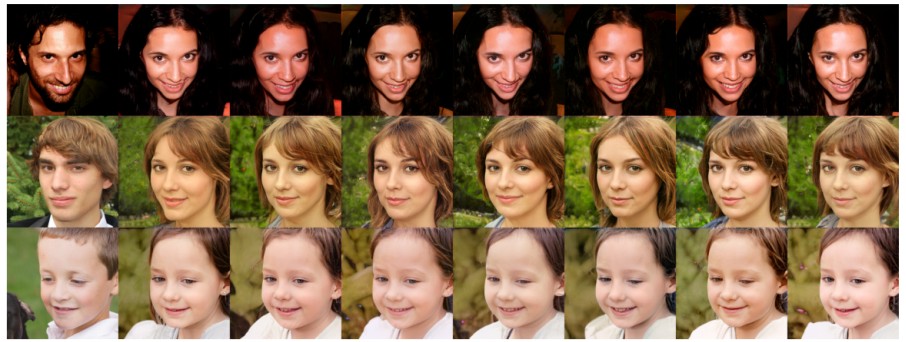

*(a)* VarEOT *Male → Female*, $\varepsilon = 0.1$. Almost no diversity.

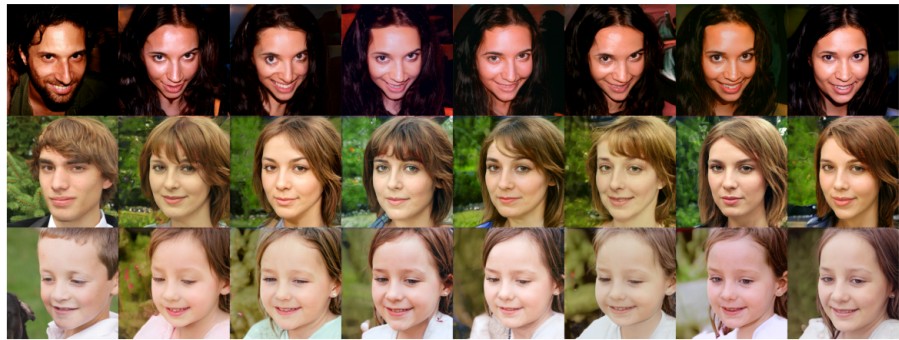

*(b)* VarEOT *Male → Female*, $\varepsilon = 0.5$. Reasonable diversity.

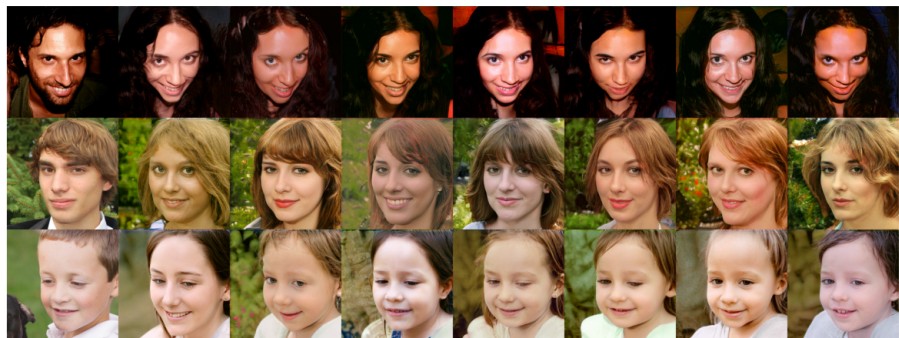

*(c)* VarEOT *Male → Female*, $\varepsilon = 1.0$. Moderate diversity.

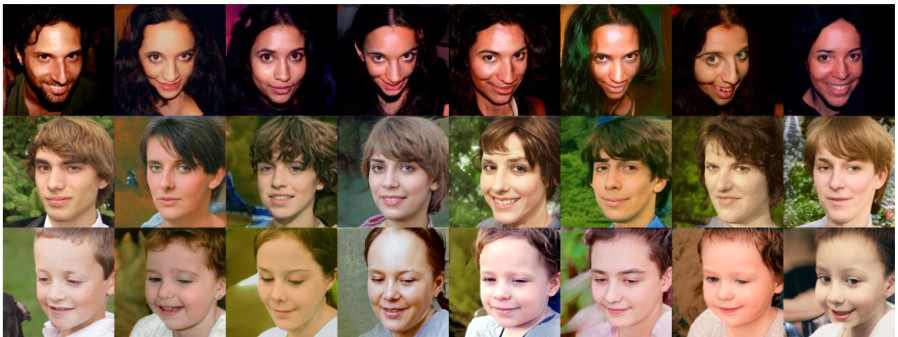

*(d)* VarEOT *Male → Female*, $\varepsilon = 10.0$. High diversity.
*Figure 6.* VarEOT with NFE=10 in task: *Man → Woman* for different $\varepsilon$

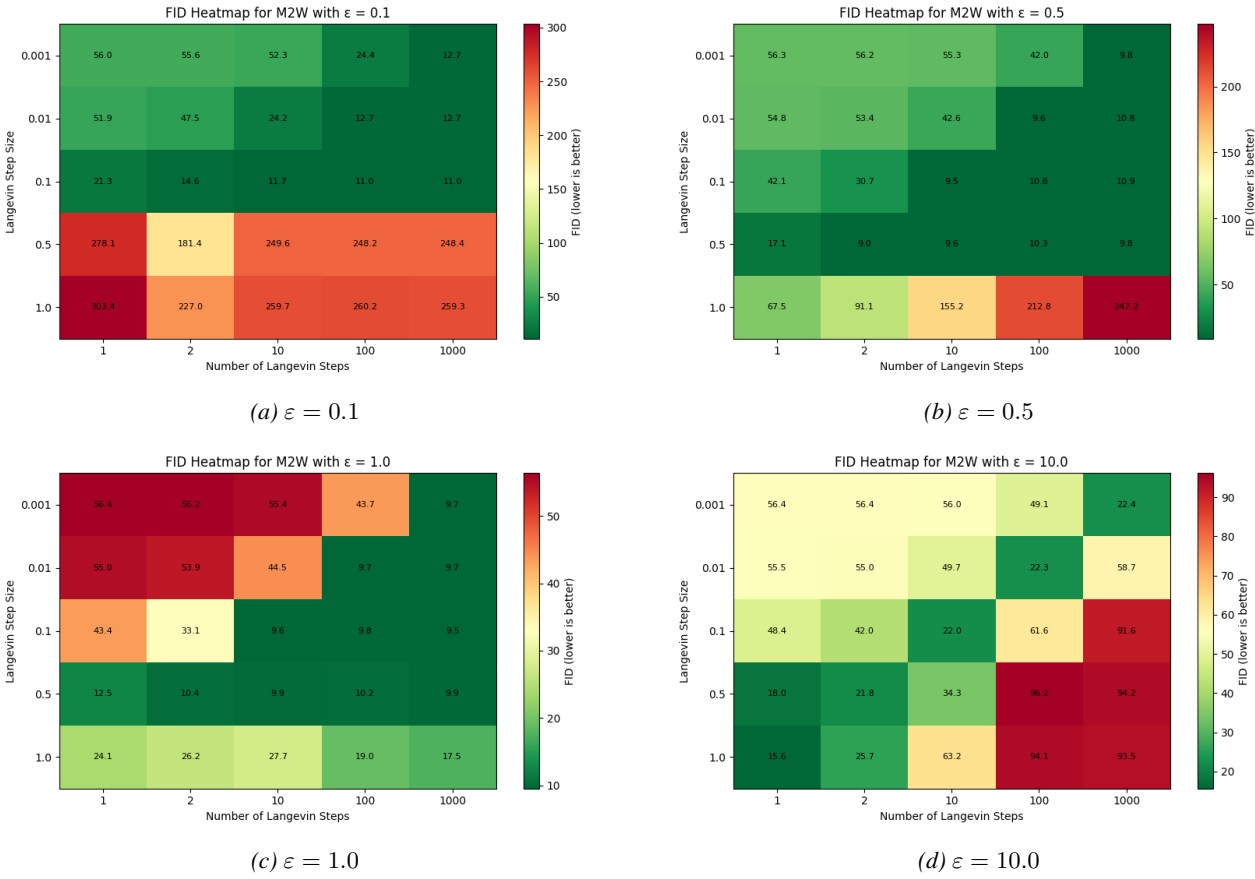

*Figure 7.* FID heatmaps for the *Male→Female* (M→F) unpaired image-to-image translation task in the ALAE latent space. Each heatmap shows the dependence of FID on the Langevin step size (rows) and the number of inference steps (columns) for a fixed value of the entropic regularization parameter $\varepsilon$. Lower values indicate better performance.

