# OpenReview forum: "Variational Entropic Optimal Transport"
_ICML.cc/2026/Conference — ICML 2026 regular_

### Official Review · Reviewer_2Z1u · 2026-03-09

**Soundness:** 3
**Presentation:** 4
**Significance:** 3
**Originality:** 3
**Overall Recommendation:** 5
**Confidence:** 5

**Summary:**

The authors propose a variational reformulation that transforms the log-partition in the EOT objective into an optimizable variable, thereby enabling neural network training without sampling.

The paper is well written and the idea of introducing variational inference in to the entropic OT problem is interesting. It overcome the similar barrier we facing in traditional probabilistic inference task that using VI to addressing the intractable posterior but in EOT task.

**Compliance With Llm Reviewing Policy:**

Affirmed.

**Key Questions For Authors:**

Can the auxiliary variable affact training stability?

How dose the choice of the entropy regularization parameter influence the quality of the learned transport map? There may be a principle way to determinate the regularization param, which would be helpful for its inference stability.

**Limitations:**

Yes

**Strengths And Weaknesses:**

The method introduce VI into EOT task, which is a new optimization formulation for EOT (to the best of reviewer's knowledge)
The VI can remove the intractable log partition in EOT so the inference can be more efficient than sampling based estimation.
The beautiful point of VI is that it can provide theoretical grantees which would be important in some critical translation tasks.



W:
The method use a minmax optimization, which can be unstable and could be sensitive to the selection of hyperparameters.

The experiment section can be improved, for example apply to realworld OT tasks to show its competitiveness and robustness

---

> ### Author Rebuttal · Authors · 2026-03-31
>
> Dear Reviewer 2Z1u, thank you for your comments. Please find below the answers to your questions and comments.
>
> **[W.1] The method use a minmax optimization, which can be unstable and could be sensitive to the selection of hyperparameters.**
>
> Thank you for the positive assessment of our variational bound derivation. We would like to note that our bound (Eq. 13, Theorem 3.2) naturally leads to a **joint maximization over both** $f$ and $\xi$:
>
> $$\text{EOT}\_{\varepsilon}(p_0, p_1) = \sup_{f, \xi}\,\mathcal{L}(f, \xi).$$
>
> Both networks $f$ and $\xi$ are trained jointly towards the same objective. We believe this point is worth highlighting more prominently in the paper, and we will do so in the updated version.
>
> **[W.2] The experiment section can be improved, for example apply to realworld OT tasks to show its competitiveness and robustness**
>
> Thank you for this valuable suggestion. Following your recommendation, we additionally evaluated VarEOT on a biological application — the analysis of single-cell data, which is important practical use case of EOT/SB methods.
>
> Specifically, we conducted experiments on the high-dimensional MSCI dataset ("Open Problems — Multimodal Single-Cell Integration"), which has been used as a benchmark by several competing methods [Tong et al., 2023; Vargas et al., 2021; Bunne et al., 2023]. The dataset consists of single-cell data from four human donors at 4 time points; we solve the EOT problem between distributions at days 2$\rightarrow$4 and evaluate the quality of the recovered transport plan using the energy distance (ED, lower is better). We consider PCA projections of DIM = 50, 100, 1000 and report the results alongside wall-clock training time in the table below.
>
> | Setup | Solver | DIM 50 | DIM 100 | DIM 1000 |
> |---|---|---|---|---|
> | Discrete EOT | Sinkhorn [Cuturi, 2013]* | 2.340 (90 s) | 2.240 (2.5 m) | 1.864 (9 m) |
> | Continuous EOT | Langevin-based [Mokrov et al., 2024, GPU V100]* | 2.39±0.06 (19 m) | 2.32±0.15 (19 m) | 1.46±0.20 (15 m) |
> | Continuous EOT | Minimax [Gushchin et al., 2023, GPU V100]* | 2.44±0.13 (43 m) | 2.24±0.13 (45 m) | 1.32±0.06 (71 m) |
> | Continuous EOT | IPF [Vargas et al., 2021, GPU V100]* | 3.14±0.27 (8 m) | 2.86±0.26 (8 m) | 2.05±0.19 (11 m) |
> | Continuous EOT | LightSB [4 CPU cores]* | 2.31±0.27 (65 s) | 2.16±0.26 (66 s) | 1.27±0.19 (146 s) |
> | Continuous EOT | **VarEOT (ours)** [GPU GTX 1080] | **2.45±0.08 (3 m)** | **2.31±0.15 (3 m)** | **1.47±0.11 (14.8 m)** |
>
> *Results marked with * are taken from the LightSB paper.*
>
> We observe that VarEOT achieves quality comparable to Langevin-based and Minimax solvers for DIM 50 and DIM 100, while being **5–15× faster** (3 m vs. 19–45 m) on a less powerfull hardware (GTX 1080 vs V100). For DIM 1000, VarEOT matches the quality of the Langevin-based solver (1.47 vs. 1.46) with comparable training time (14.8 m vs. 15 m). We will include these results in the updated version of the paper. We hope this additional experiment addresses your concern.
>
> **[Q.1] Can the auxiliary variable affact training stability?**
>
> We note that in the single-cell experiments described in **[W.2]** the final confidence intervals of our method are narrow and similar to those of LightSB and EgNOT, which do not use auxiliary variables to approximate the loss we optimize. Thus, we conclude it does not introduce training instability.
>
> **[Q.2] How dose the choice of the entropy regularization parameter influence the quality of the learned transport map? There may be a principle way to determinate the regularization param, which would be helpful for its inference stability.**
>
> Entropy regularization parameter $\epsilon$ determines the stochasticity of the learned map. Specifically, for $\varepsilon = 0$ the solution is a deterministic OT map, with larger $\varepsilon$ it becomes more stochastic, and in the limit $\varepsilon \rightarrow \infty$ it transforms into an independent plan $\pi(x, y) = p(x)p(y)$.
>
> Specific values of the coefficient $\varepsilon$ fully depend on the particular data distributions $p(x_0)$ and $p(x_1)$, because they determine the magnitude of the cost part (first term) in the EOT objective:
> $$
> \min\_{\pi \in \Pi(p\_0,p\_1)}  \left[ \mathbb{E}\_{(x\_0,x\_1)\sim \pi} \left[\frac{\|x\_0-x\_1\|\^2}{2}\right] - \varepsilon \int\_{\mathbb{R}^D} H\bigl(\pi(\cdot \mid x\_0)\bigr)\, p\_0(x_0)\, dx\_0 \right]
> $$
> For instance, scaling distributions by 2 times will lead to the necessity to scale the coefficient $\epsilon$ by 4 times to keep the same ratio between cost and regularization terms.
>
> Thus, there is general guidance on how one can tune the parameters towards a more stochastic or a more deterministic way, but the absolute value highly depends on the particular datasets.
>
> **Concluding remarks**. We would be grateful if you could let us know if our explanations have been satisfactory. We are also open to discussing any other questions you may have.

---

> > ### Author Rebuttal · Reviewer_2Z1u · 2026-04-03
> >
> > Thanks for the detailed response; I keep my positive score.

---

### Official Review · Reviewer_vf1j · 2026-03-12

**Soundness:** 4
**Presentation:** 3
**Significance:** 3
**Originality:** 3
**Overall Recommendation:** 4
**Confidence:** 3

**Summary:**

The paper considers continuous entropic OT with quadratic cost between absolutely continuous source and target measures p_0,p_1, in the “weak dual / semi-dual” formulation with a single potential f. The key difficulty in prior neural weak-dual solvers is handling the log-partition term log⁡Z(f,x_0), which is intractable to compute and differentiate exactly in high dimensions.

The main idea is to introduce an auxiliary scalar function ξ(x_0) and apply a simple exponential-tilting inequality to obtain a variational upper bound on log⁡Z(f,x_0). Plugging this into the weak dual objective yields a new “variational dual” objective L(f,ξ) that can be estimated by Monte Carlo using only samples from p_0,p_1 and a standard Gaussian, without sampling from the model transport plan.

The authors propose VarEOT: parameterize both f and ξ with neural networks, maximize an empirical estimate of L(f,ξ) using stochastic gradients, and at test time recover the conditional EOT plan π(x_1∣x_0) from the learned potential f_θ and sample from it via Langevin dynamics. They provide (i) a KL-type excess-risk bound linking suboptimality of L(f,ξ) to KL divergence between the learned plan and the true EOT plan, and (ii) finite-sample rates for the estimation error plus an approximation error bound under universal approximation by neural nets.

Empirically, the authors show 2D Gaussian→Swiss-roll examples (illustrating the effect of the regularization) and unpaired image-to-image translation in ALAE’s latent space on FFHQ for several domain splits, comparing primarily to EgNOT and LightSB across a range of ε and Langevin budgets. On these benchmarks, VarEOT typically matches or improves FID/LPIPS relative to both baselines, while enjoying simulation-free training.

**Compliance With Llm Reviewing Policy:**

Affirmed.

**Key Questions For Authors:**

1. Clarify theoretical positioning. Explicitly relate the error decomposition and rates to neural EOT estimation work (e.g. ) and to existing statistical bounds for entropic OT, emphasizing what is new.

2. Add at least one more dataset or task family. Even a second image domain (AFHQ, ImageNet subset) or a non-image continuous dataset would broaden the empirical message.

3. Include runtime / training-cost comparisons. Since “simulation-free training” is central, a wall-clock and GPU-step comparison to EgNOT and LightSB would materially strengthen the significance claim.

4. Discuss limitations more concretely. In particular, (i) variance / stability of the exponential terms, (ii) sensitivity to ξ’s capacity, and (iii) restriction to quadratic cost and Gaussian kernels.

**Limitations:**

The paper acknowledges some limitations, however, the discussion can be improved following the suggestions specified in "Key Questions For Authors" above.

**Strengths And Weaknesses:**

The technical development appears to be solid and well executed.

The paper is generally well written and structured. The notation is consistent, and the high-level motivation is well explained: weak-dual EOT is attractive but the log-partition is painful; VarEOT resolves this tension without sacrificing expressiveness.
The related-work section is thorough on weak-dual EOT and SB-based methods and correctly distinguishes continuous OT from discrete OT approaches that do not directly address out-of-sample conditional plans. Adding explicit discussion of neural semi-dual EOT estimators and other recent simulation-free entropic OT variants would improve the positioning.

Compared to EgNOT, the new solver retains the appeal of a flexible, neural weak-dual potential but removes the need to run MCMC during training. This directly addresses a known practical drawback of EgNOT, namely the training-time sampling cost and its sensitivity to MCMC hyperparameters.
Compared to LightSB, VarEOT avoids the explicit Gaussian-mixture parameterization of the conditional plan and the resulting structural constraints; the conditional is instead induced by a general neural potential as in EgNOT. In this sense, the method does occupy the intended “middle point” between the two prior approaches (simulation-free training and unrestricted potential parameterization), at least for the quadratic-cost, continuous EOT setting the paper focuses on.

Section 3.3 develops an error decomposition for the discrepancy between the learned coupling and the true EOT plan in terms of: (i) an estimation/generalization error and (ii) an approximation error arising from restricting to neural network classes. The estimation error is bounded in terms of Rademacher complexities under assumptions of compact supports and Lipschitz-bounded potentials. The approximation error is handled via a universal-approximation-style argument for increasing families of neural-network classes, showing that, for sufficiently expressive networks, the supremum of the variational objective over these classes can be made arbitrarily close to the optimal weak dual value. In combination, these results provide control for the KL divergence between the learned and optimal plans. The arguments used in the paper are closely aligned with existing statistical analyses for entropic OT and related problems (e.g., works on entropic barycenters and neural EOT estimation), and the paper acknowledges this connection in the discussion of prior work. The specific KL link for the plan under this new variational objective appears to be a reasonably natural but useful refinement rather than a radical departure from existing theory.

In the ALAE/FFHQ setup of the empirical evaluation, VarEOT is compared against EgNOT and LightSB both qualitatively and quantitatively  across several regularization strengths and inference-time Langevin budgets. The reported numbers in Table 2 indicate that VarEOT often matches or improves upon these baselines on these metrics, particularly when considering reasonable numbers of Langevin steps. Sensitivity analyses for ε, step sizes, and the number of inference steps provide additional insight into how the method behaves in practice.
At the same time, the empirical scope is concentrated on a single high-dimensional domain (ALAE/FFHQ), and the current draft does not include explicit runtime comparisons, even though simulation-free training is a central motivation. Additional datasets or modalities, and explicit training-time comparisons to EgNOT and LightSB, would strengthen the empirical case, particularly for a broad ML audience.
Overall, the empirical section is competent but somewhat narrow.

---

> ### Author Rebuttal · Authors · 2026-03-31
>
> Dear Reviewer vf1j, thank you for your comments. Please find below the answers to your questions and comments.
>
> **Q1. Theoretical positioning compared to existing EOT estimation work/existing statistical bounds for entropic OT**
>
> The situation with EOT estimation/stat. bounds - related research is as follows. Some works, e.g., [Genevay, Rigollet, Mena], study EOT from the statistical learning perspectives, but focus primarily on the EOT **cost** value estimation or the accuracy of the recovered barycentric projection $x \mapsto \int y d \hat{\pi}(y \vert x)$ in the **non-parametric** setup. In contrast, we deal with statistical properties of recovered **plan** $\pi^{\hat{f}}$ in **parametric** setup ($f$ are NNs).
>
> There are only few works, which consider similar theoretical setup $-$ [EgNOT] and [EgBary] (mentioned in "Relation to prior work" paragraph, lines 321-329). Below we provide the direct comparison.
>
> 1. **[EgNOT]** Similar to ours, they rely on a semi-dual EOT objective and split the discrepancy between GT and recovered plans $\pi^*$ and $\pi^{\hat{f}}$ into estimation and approximation errors; the estimation error is then bounded via Rademacher complexities of certain function classes.
>
>     In contrast, our VarEOT provides a deeper analysis. We also decompose the discrepancy, though this is more involved due to the auxiliary potential $\xi$, and then **analyze each component separately**. We derive explicit rates (in terms of training data sizes' $N$ and $M$) for the estimation error (Thm. 3.5) and prove a universal approximation guarantee for approximation error (Thm. 3.7). Overall, section 3.3 establishes statistical consistency of our solver, unlike the theory in EgNOT.
>
> 2. **[EgBary]** They perform accurate analysis of their objective (with $N$ and $M$ - dependent rates and universal approximation guarantees) but for a different problem $-$ EOT barycenter. Actually, we adopt certain ideas from EgBary in our proofs, yet we think it is a common and good practice in science.
>
> [Genevay] Genevay et. al., Entropy-regularized optimal transport for machine learning, PhD thesis'19
>
> [Rigollet] Rigollet et. al., On the sample complexity of entropic optimal transport, Ann. Statist.'25
>
> [Mena] Mena et. al., Statistical bounds for entropic optimal transport: sample complexity and the central limit theorem, NeurIPS'19
>
> [EgNOT] Mokrov et. al., Energy-guided Entropic Neural Optimal Transport, ICLR'24;
>
> [EgBary] Kolesov et. al., Energy-Guided Continuous Entropic Barycenter Estimation for General Costs, NeurIPS'24;
>
> **[Q.2] Add at least one more dataset or task family. Even a second image domain (AFHQ, ImageNet subset) or a non-image continuous dataset would broaden the empirical message.**
>
> Following your recommendation, we additionally evaluated VarEOT on a popular OT biological application — the analysis of single-cell data, which is an important practical use case of EOT/SB methods. **Please see the answer to weakness W2 of the reviewer 2Z1u.**
>
> **[Q.3] Include runtime / training-cost comparisons. Since “simulation-free training” is central, a wall-clock and GPU-step comparison to EgNOT and LightSB would materially strengthen the significance claim**
>
> Thank you for this important question. Following your suggestion, we conducted wall-clock time measurements. **Please see the answer to question Q2 for the reviewer 2CUN**
>
> **[Q.4] Discuss limitations more concretely. In particular, (i) variance / stability of the exponential terms, (ii) sensitivity to ξ’s capacity, and (iii) restriction to quadratic cost and Gaussian kernels.**
>
> We agree with the raised concerns and will add an extended limitations discussion for the final version of the paper, addressing each point:
>
> **(i) Variance / stability of the exponential terms.** We acknowledge that the exponential terms in the objective may cause numerical instability during training. In our implementation, we address this via clipping large values of the exponential terms before computing gradients, as well as gradient clipping, and we will explicitly mention this in the updated text.
>
> **(ii) Sensitivity to $\xi$'s capacity.** We agree that the quality of optimization, by apply real world the expressiveness of the auxiliary network $\xi$, and that the choice of its architecture remains a practical question. We note that in our experiments a standard MLP for $\xi$ proved sufficient, and we will acknowledge this dependence explicitly in the limitations section.
>
> **(iii) Restriction to quadratic cost and Gaussian kernels.** We acknowledge that our current framework is specialized to the quadratic cost (Gaussian kernel), and we will highlight extending to general costs as a promising direction for future work.
>
> **Concluding remarks**. We would be grateful if you could let us know if our explanations have been satisfactory. If so, we kindly ask that you consider increasing your rating. We are also open to discussing any other questions you may have.

---

> > ### Author Rebuttal · Reviewer_vf1j · 2026-03-31
> >
> > I thank the authors for their response and additional analysis, which have increased my confidence in my positive original review. I have increased some of the original scores as a result of the authors response. The paper is still specialized rather than broad, but within its area it is well executed.

---

### Official Review · Reviewer_2CUN · 2026-03-13

**Soundness:** 3
**Presentation:** 3
**Significance:** 3
**Originality:** 3
**Overall Recommendation:** 4
**Confidence:** 3

**Summary:**

This paper proposes a variational reformulation of entropic optimal transport. From a nice reformulation, the proposed algorithm enables simulation-free training with flexible neural parameterization. The paper also provides theoretical guarantees and experiments on toy problems and image translation.

**Compliance With Llm Reviewing Policy:**

Affirmed.

**Final Justification:**

My questions have been well addressed. I would be happy to maintain my score (4).

**Key Questions For Authors:**

1. Theorem 3.5 gives a bound in terms of N and M. Can this result be used to guide experiments or improve practical performance? Please also provide how other theories are connected to experiments
2. In Table 2, performance improves significantly when NFE is large, but is weaker for small NFE in some settings. Can the authors clarify this phenomenon?
3. Can the authors compare the time complexity of the proposed model with that of other existing models?

**Limitations:**

Yes

**Strengths And Weaknesses:**

I found the formulation interesting and theoretically well-founded. The method addresses a meaningful limitation of prior weak-dual EOT approaches, and the paper is generally clear and technically solid. My main concern is that the connection between theory and experiments is not fully clear. In particular, Theorem 3.5 provides an upper bound in terms of the sample sizes N and M, but the experiments mainly focus on performance across different inference budgets rather than on how varying N and M affect the results. Because of this, it is hard to see how the theorem informs practical experimental design or performance.

---

> ### Author Rebuttal · Authors · 2026-03-30
>
> Dear Reviewer 2CUN, thank you for your comments. Please find below the answers to your questions and comments.
>
> **[W1, Q1]: "[...] connection between theory and experiments is not fully clear." Theorem 3.5 (bound in terms of $N$ and $M$)/other theories vs. practical experimental design**
>
> Let us clarify the positioning of our Theorem 3.5 (and other theoretical results in section 3.3) - what is the role of this theorem in our manuscript, and why it is important and worth presenting.
>
> In fact, Theorem 3.5 could not be directly "estimated" in practice, since term $\mathbb{E} \sup_{f, \xi} \vert \mathcal{L}(f, \xi) - \widehat{\mathcal{L}}(f, \xi) \vert$ could not be directly computed (we could not iterate over all NN parameters' choices to find the $\sup$). In this sense, our Theorem 3.5 indeed does not provide any practical "guidance" and could not be directly used to monitor the practical performance.
>
> However, Theorem 3.5 is a **necessary ingredient** to make the final conclusion of \S 3.3 (see "Summary" paragraph, lines 315-319): if we have expressive enough Neural Networks + large enough data then we can recover the optimal OT plan $\pi^*$ sufficiently good (our Proposition 3.4 in tandem with our Theorems 3.5 and 3.7). Scientifically speaking, our developed methodology for solving EOT is **statistically consistent** - this is an important property that, ideally, any method used to solve an optimal transportation problem should satisfy.
>
> **[Q2] In Table 2, performance improves significantly when NFE is large, but is weaker for small NFE in some settings. Can the authors clarify this phenomenon?**
>
> At the inference stage, we use Langevin sampling (Algorithm 2), which provides convergence guarantees to the target distribution in the limit of infinite NFE ($S \rightarrow \infty$) and infinitely small step size ($\eta \rightarrow 0$). Thus, these experimental results align with the theoretical perspective.
>
> **[Q3] Can the authors compare the time complexity of the proposed model with that of other existing models?**
>
> Thank you for this important question. Following your suggestion, we conducted wall-clock time measurements on the M $\rightarrow$F experiment ($\varepsilon{=}1.0$) using the same hyperparameters as in the original paper for all solvers. The results are presented in the table below.
>
> | Method | Config | Time | Iter | FID ↓ |
> |---|---|---|---|---|
> | **VarEOT (ours)** | K=512 | 409 s | 10000 | **9.575** |
> | **VarEOT (ours)** | K=256 | 215 s | 10000 | **10.55** |
> | **VarEOT (ours)** | K=128 | 131 s | 10000 | **11.43** |
> | **VarEOT (ours)** | K=64 | 85 s | 10000 | **12.85** |
> | **VarEOT (ours)** | K=32 | 85 s | 10000 | **13.45** |
> | LightSB | — | 191 s | 10000 | 22.63 |
> | EgNOT | — | 1692 s | 10000 | 17.58 |
>
> We observe that the training time of VarEOT scales with the number of particles $K$: larger $K$ leads to longer training, but also to better FID. Importantly, VarEOT achieves superior FID compared to both LightSB and EgNOT across all considered values of $K$, while requiring significantly less training time than EgNOT (e.g., 131 s vs. 1692 s for comparable or better quality). We attribute this to our simulation-free training objective, which avoids costly trajectory simulation required by EgNOT. We will include these results in the updated version of the paper.
>
> **Concluding remarks**. We would be grateful if you could let us know if our explanations have been satisfactory. If so, we kindly ask that you consider increasing your rating. We are also open to discussing any other questions you may have.

---

> > ### Author Rebuttal · Reviewer_2CUN · 2026-04-03
> >
> > Thank you to the authors for their response. I read the rebuttal carefully, and my questions have been well addressed. In particular, I appreciate the additional table, which was helpful. I would be happy to maintain my score.

---

### Decision · Program_Chairs · 2026-04-30

**Decision:**

Accept (regular)

**Comment:**

This paper studies entropic optimal transport in continuous spaces with quadratic cost and proposes VarEOT, a variational reformulation of the weak semi-dual objective that removes the intractable log-partition term by introducing an auxiliary positive normalizer, leading to a simulation-free training objective compatible with flexible neural parameterizations. The main contributions are this reformulation, accompanying theoretical guarantees on approximation and generalization, and empirical validation on synthetic data, unpaired image translation, and an additional single-cell application added after rebuttal. The main strengths are that the method addresses a genuine bottleneck in neural weak-dual EOT, is technically well supported, and offers an appealing practical advantage through simulation-free training while remaining competitive empirically. The main weaknesses are that the empirical scope is still somewhat specialized and that the practical implications of the theory and the limitations of the quadratic-cost setting were not fully clear in the original submission; the rebuttal addressed these points reasonably well by adding runtime and single-cell experiments, clarifying the purpose of the theory, and acknowledging the method’s practical caveats, though the overall scope remains somewhat narrower than the broader methodological framing might suggest. Overall, the paper presents a sound and useful contribution with positive reviewer support, and I recommend acceptance.